# Fast capillary waves on an underwater superhydrophobic surface

Maxime Fauconnier ●[1] ✉, Bhuvaneshwari Karunakaran ●[2], Alex Drago-González ●[1], William S. Y. Wong ●[2], Robin H. A. Ras ●[2] ✉ & Heikki J. Nieminen ●[1] ✉

The propagation of interfacial waves in free and constrained conditions, such as deep and shallow water, has been broadly studied over centuries. It is a common event that anyone can witness, while contemplating the ocean waves washing ashore. As a complementary configuration, this work introduces waves propagating on an interface restricted by its pinning to the solid microstructures of an underwater superhydrophobic surface. The latter has the ability to stabilize a well-defined microscale gas layer, called a plastron, trapped between the water and the solid phase. The acoustic radiation force produced with focused MHz ultrasound successfully triggers kHz "plastronic waves", i.e., capillary waves travelling on a plastron's gas-water interface. The exposed waves possess interesting features, i.e., (i) a high propagation speed up to 45 times faster than conventional deep water capillary waves of comparable wavelength and (ii) a relation of the propagation speed with the geometry of the microstructures. Based on this and on the observed variation of wave speed over time in conditions of gas-undersaturated or -supersaturated water, the usefulness of the plastronic waves for the non-destructive monitoring of the plastron's stability and the spontaneous air diffusion is eventually demonstrated.

Capillary waves, also known as ripples, are surface tension-dominated mechanical perturbations of a liquid-gas interface. Evocations date back at least to the first century AD, when Pliny the Elder was reporting on the absence of ripples in the wake of ships releasing surfactants[1]. The earliest description of capillary waves in the modern scientific sense is attributed to John Scott Russell[2]. His investigation portrayed the orbital motion of the water molecules carried on by the swelling interface and measured propagation speeds in the range 0.2–0.9 m s$^{-1}$. The influence of a nearby seabed on the wave motion has been first assessed for the case of gravity-driven waves with longer wavelength $\lambda$, in a configuration of intermediate depth of the water medium, with a first linear model proposed by Sir Airy in 1841[3], further developed to account for weak nonlinear effects by Sir Stokes[4] in 1847. Later extensions to these models would consider e.g., high wave steepness[5]

and pure capillary waves[6]. Together, these works would contribute in particular to describing a wavenumber-to-frequency relation, commonly called dispersion relation. For a wave travelling at an interface between two fluid phases (i.e., an interfacial wave) $a$ and $b$, the angular frequency $\Omega(k)$ writes

$$\Omega^2(k) = \frac{gk\,|\,\rho_a - \rho_b\,| + \sigma k^3}{\rho_a \coth(kh_a) + \rho_b \coth(kh_b)}, \tag{1}$$

where $g$ and $\sigma$ are gravity and surface tension. $k = 2\pi/\lambda$ is the wavenumber. $\rho_a$ and $h_a$ (respectively, $\rho_b$ and $h_b$) are the density and thickness of the fluid phase $a$ (respectively, $b$). At an air-water interface, a minimum for the wave propagation speed, here referred to as the phase speed $c_p = \Omega(k)/k$, can be found for $\lambda \simeq 1.7$ cm and equals $ca.$

[1]Medical Ultrasonics Laboratory (MEDUSA), Department of Neuroscience and Biomedical Engineering, Aalto University, Espoo, Finland. [2]Department of Applied Physics, Aalto University, Espoo, Finland. ✉e-mail: maxime.fauconnier@aalto.fi; robin.ras@aalto.fi; heikki.j.nieminen@aalto.fi

0.23 m s⁻¹, in a condition of deep water ($h \gg \lambda$). Interfacial waves on water are generally categorised, according to their wavelength into two distinct groups, capillary ($\lambda \leq 1.7$ cm) and gravity ($\lambda \geq 1.7$ cm) waves, mainly governed by distinct restoring forces, respectively surface tension and gravity[7]. Capillary and gravity waves are dispersive, meaning that their propagation speed depends on their wavelength, according to Eq. (1), a graphical representation of which is provided as Supplementary Information (SI), in Fig. S1.

The wave-driven orbital motion of water molecules constrained by a shallow water configuration ($h \ll \lambda$) experiences a no-slip boundary condition at the water-bottom interface[8], causing the wave to slow down[3] and its amplitude to increase[9]. This effect is known as wave shoaling and is opposed to the restoring forces, such as gravity and interfacial tensions, which try to restore the equilibrium state. The pinning contact of the water interface to the walls of a channel in which a wave travels can also exert an additional restoring force, opposite to the interface displacement, which further stiffens the gas-water interface[10] and speeds up the wave propagation[10,11]. This effect is intensified, when the wall spacing is narrowed[12] and the water interface becomes curved[13]. In contrast, a condition of freely moving contact line agrees with the theory of waves on constraint-free interfaces[10,14]. Together, this abundant literature constitutes in-depth knowledge of water waves in various configurations, which all share a common determinant, namely the assumption that the air or gas phase above the water interface is boundary-free and inviscid.

While capillary waves have been extensively studied in free conditions (e.g., deep water)[2–7] and in constrained conditions (e.g., shallow water or narrow channel)[3,8–14], to the best of our knowledge, systematic studies on capillary waves at an interface constrained by its pinning to an array of microstructures are still lacking. Such conditions could be facilitated by the superhydrophobic state exhibiting a superior water repellency. Nature itself offers fascinating examples, such as lotus[15] and salvinia[16] leaves, aquatic arthropod exoskeletons[17], feathers[18], serving as remarkable sources of inspiration[19]. Superhydrophobicity is enabled by a combination of surface chemistry[20] and nano-micro topography, which prevents liquid entry and impalement[20,21]. Underwater immersion, a thin layer of air is trapped between the micropillars, where the micropillar tops support the water like a fakir on a bed of nails. Collectively, this is known as a plastron, or the Cassie state[22]. However, the water-repellent feature can be compromised by spontaneous changes in the environment[20]. In particular, the plastron can experience depletion as a result of gas dissolution or pressure-induced impalement[23,24]. Eventually, the plastron could be lost and the surface fully wetted, the so-called Wenzel state[25], while partially wet states would be here referred to as intermediate states. The plastron condition is typically assessed optically by bright-field microscopy[23,26], total internal reflection technique[24,27], or confocal microscopy[28]. Acoustics-based alternatives to optics[21] show limitations due to (i) complexity and cost related to integrating a transducer into the studied surface[29,30], or (ii) lack of access to the plastron dynamics and intermediate wetting states[31].

Mechanical waves carry their propagation medium's properties, making them useful tools for investigating optically opaque or contamination-sensitive media, such as, respectively, the Earth's crust[32] and biological tissues[33], or soft matter and biological fluids[34,35]. With this in mind, the examination of the dynamics of the gas-water interface, constrained by microstructures, in response to mechanical stress could provide a means for remotely monitoring the plastron's condition, wetting state, and spontaneous gas diffusion over time. Accordingly, this work presents an original configuration of capillary waves induced by ultrasound (US), travelling on a gas-water interface bounded to the solid microstructures of a superhydrophobic plastron. The wave behaviour is studied by varying the density of the three-phase contact line and the pillar height. The sensing potential of these waves for remotely monitoring the plastron and its intermediate wetting state is eventually addressed.

## Results and discussion

### Superhydrophobic underwater medium for interfacial waves

Made of PDMS (polydimethylsiloxane), superhydrophobic samples with well-defined surface roughness were prepared by soft lithography, the details of which are given in "Microfabrication". The surface topography consists of a 5 mm × 5 mm array of cylindrical micropillars arranged in a square lattice. While the pillar diameter is kept constant (20 μm), the investigated configurations differ by the pillar height $h$ (21, 25, 37, 53 and 71 μm) or the inter-pillar spacing $s$ (15, 20, 25, 35, 45, 55 and 65 μm). A graphical summary of the employed configurations is provided in SI, Fig. S2. The experimental setup, schematically illustrated in Fig. 1a, b and described in "Experimental setup", employs a superhydrophobic surface immersed in water. The image of a specimen ($h = 25$ μm, $s = 25$ μm), acquired by Scanning Electron Microscopy, is given in Fig. 1c. Along with high-speed imaging (159090 fps) and a high-intensity focused US (HIFU) transducer (2.5 MHz), the experimental configuration allowed to generate and to observe the periodic perturbations of the gas-water interface of a plastron induced by acoustic radiation force (ARF), enabled by focused US. Given that the imaging frame rate is considerably smaller than the acoustic frequency (2.5 MHz), this work does not report on the acoustic time-scale dynamics of the plastron, but rather on the fluidic time-scale phenomena set into motion by the second-order nonlinear forces of the acoustic field. The ARF-induced interface motion modifies the light refraction leading to local changes of light transmission through the sample, providing a top-view visualisation of the plastron spatiotemporal perturbations at microsecond time scales.

### Acoustically induced plastronic waves

Periodically poking an underwater plastron with an US excitation with spatial acoustic distribution given in Fig. 1e allows to generate coherent ripples in a circularly symmetric fashion, as exemplified in Fig. 1f. This essentially results from the acoustic waves having sufficient time-averaged energy density to provide an ARF able to deform the gas-water interface[36]. As per existing theoretical models[37], a focal acoustic pressure of 1.5 MPa (measurement in a free field condition) is expected to be able to induce a Cassie-to-Wenzel transition of the plastron, given that this pressure would correspond to a radiation force ≥540 Pa, which is superior to the critical impalement pressure of all plastron configurations investigated in this work. However, considering that the US actuation employed here is of short duration (<1 ms) and modulated in amplitude, as detailed hereafter, it might not provide sufficient time for the inertia-driven interface dynamics to locally generate plastron collapse. This aspect is briefly commented in SI, Section 1.A.

Since conventional interfacial waves are dispersive, a control of the wave frequency is needed to assess the influence of the plastron geometry only on the wave behaviour. To achieve this, an amplitude modulation (AM) was applied to the 2.5 MHz sinusoidal wave delivered by the HIFU transducer, expected to yield an ARF frequency acting on the superhydrophobic surface approximately at double the frequency of the AM. The employed driving signal has the form $y(t) = A \sin(2\pi f_{ac} t) \cdot \sin(2\pi t / T_{AM})$, with the constants $A = 0.5$ V and $f_{ac} = 2.5$ MHz. The only changing parameter in the signal is the AM period $T_{AM}$, which is in the range 100–500 μs and affects the total duration of the signal that lasts 3 cycles of AM, so that $t \in [0, 3T_{AM}]$.

Under the action of that AM driving pressure, a dynamic game of forces sets up between the US waves periodically poking the interface and the interfacial tensions trying to restore its equilibrium. This results in the generation of capillary waves travelling at the gas-water interface of the plastron, therefore referred to in this study as plastronic waves. A video recording can be found in SI (Movie 1). In the exemplary case of Fig. 1f, $T_{AM} \cong 160$ μs, and the signal duration is thus $3T_{AM} \cong 480$ μs. Three cycles of AM appeared to be a good compromise of duration considering that a longer excitation may result in interference phenomena of the interfacial waves due to reflection from the

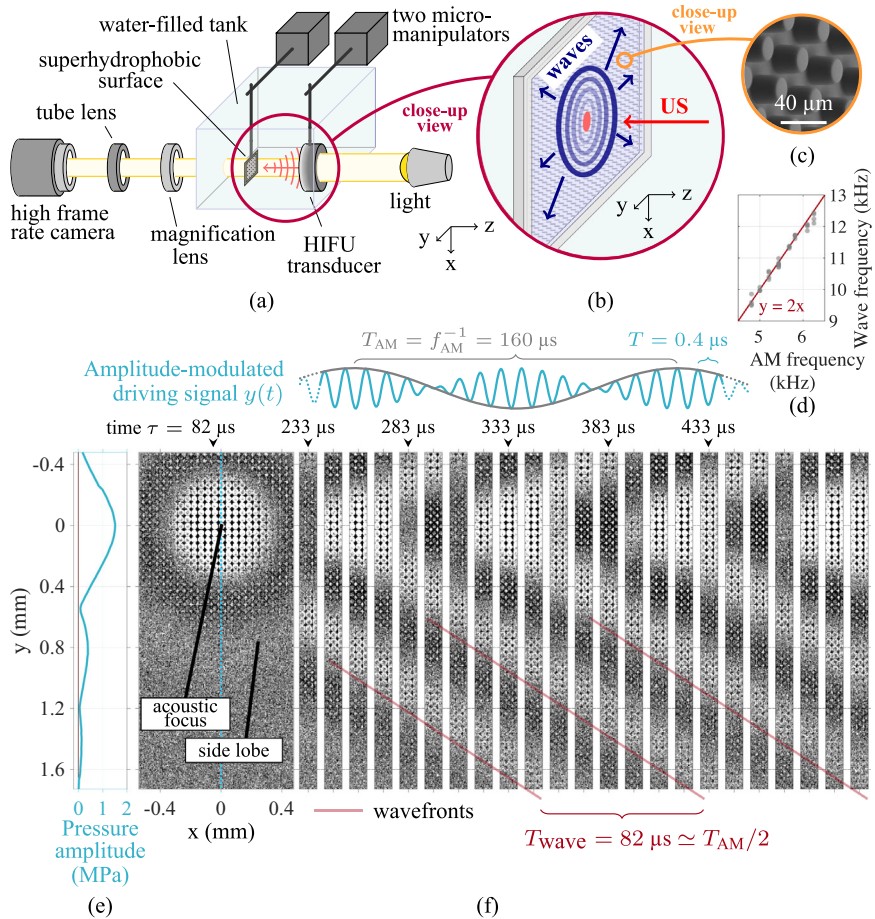

**Fig. 1 | Experimental method. a** The experimental setup, schematised here, featuring a high-speed camera (159090 fps) and a 2.5 MHz high-intensity focused US (HIFU) transducer, allows the study of a water-submerged superhydrophobic surface placed at the focus of co-axial optical and acoustic fields. The response of the plastron to an amplitude-modulated (AM) US pulse takes the shape of interfacial perturbations travelling along the gas-water interface from the acoustic focal point, the ripples with same coherence forming patterns of concentric circles, schematised in (**b**). **c** An exemplary image acquired by Scanning Electron Microscopy shows the details of a superhydrophobic sample with pillar spacing $s = 25\,\mu m$ and height $h = 25\,\mu m$. **d** For the same pillar geometry, the wave frequency as a function of the applied AM frequency always exhibits a 2:1 ratio, demonstrating the control of the frequency of the produced plastronic waves. **e** The axisymmetric field of acoustic pressure produced by the HIFU transducer was measured in a free field (in the absence of a superhydrophobic surface) using a needle hydrophone. **f** The radial waves at the gas-water interface ($s = 25\,\mu m$, $h = 25\,\mu m$) are optically assessed from the top-view refraction patterns induced by the moving gas-water interface. The wave analysis based on the tracking of the wavefronts, i.e., the wave lines of same phase, normally done in polar coordinates (Details in SI, Figs. S10 and S11), is here conceptualised on an unprocessed high-speed footage.

outer boundary of the superhydrophobic sample. On the other hand, a shorter excitation did not allow the generation of enough wavefronts to provide a sensitive detection of the wave properties, such as phase speed, wavelength, frequency, and attenuation ratio, in the employed optical arrangement. The details of the generation process of the plastronic waves is outlined in the following.

The time instant $t = 0\,\mu s$ corresponds to the moment, when the generation of a single AM US pulse $y(t)$ is initiated. Because the HIFU transducer's focusing shape has a nominal 50 mm radius of curvature, the acoustic perturbation has to propagate from the transducer for about $34\,\mu s$, theoretically, before it reaches the gas-water interface of the plastron, placed at the acoustic focus of the transducer. This estimation considers a speed of sound in water $= 1490\,m\,s^{-1}$ at 25 °C. When the US pulse should have started to impact the plastron, a pattern of concentric circles emerges from stillness. The optical contrast reaches a maximum around time $t = 82\,\mu s$, as illustrated in the first snapshot in Fig. 1f. In this exemplary case, $82\,\mu s$ approximately equals $34\,\mu s + T_{AM}/4 = 74\,\mu s$, which should temporally correspond to the first maximum of AM, and, therefore, to a maximum of acoustic energy. The bright areas of this pattern spatially correlate with the experimentally measured locations of both the acoustic

focus and the first side lobe, the latter located at a radial distance $r \cong 0.8\,mm$ from the acoustic epicentre, as shown in Fig. 1e and labelled in Fig. 1f. It can therefore be deduced that brighter and darker areas respectively transcribe troughs (inward excursion, i.e., towards the gas phase) and crests (outward excursion, i.e., towards the water) of the gas-water interface, producing a pattern of optical refraction, i.e., the consequence of transmitted light bending due to the deformation of the gas-water interface. This is further confirmed and elaborated thanks to a side-view demonstration of plastron deformation performed in a superhydrophobic micro-channel, as discussed in Section 1.A of the SI and illustrated with experimental images in Fig. S3.

In Fig. 1d, the frequency of the plastronic waves is studied as a function of the signal AM frequency. When the latter varies in the range 4.8–6.3 kHz, the frequency of the plastronic waves varies in the range 9.5–12.1 kHz. A ratio 2:1 of the wave frequency to the AM frequency is thus repeatedly observed. This is true for any micropillar geometry, as shown in Fig. S4, confirming that the AM ARF poking the plastron with periodicity $T_{AM}/2$ is the driving mechanism of the plastronic waves. The determination of the range of AM frequencies employed for each microstructure geometry results from experimentation on the

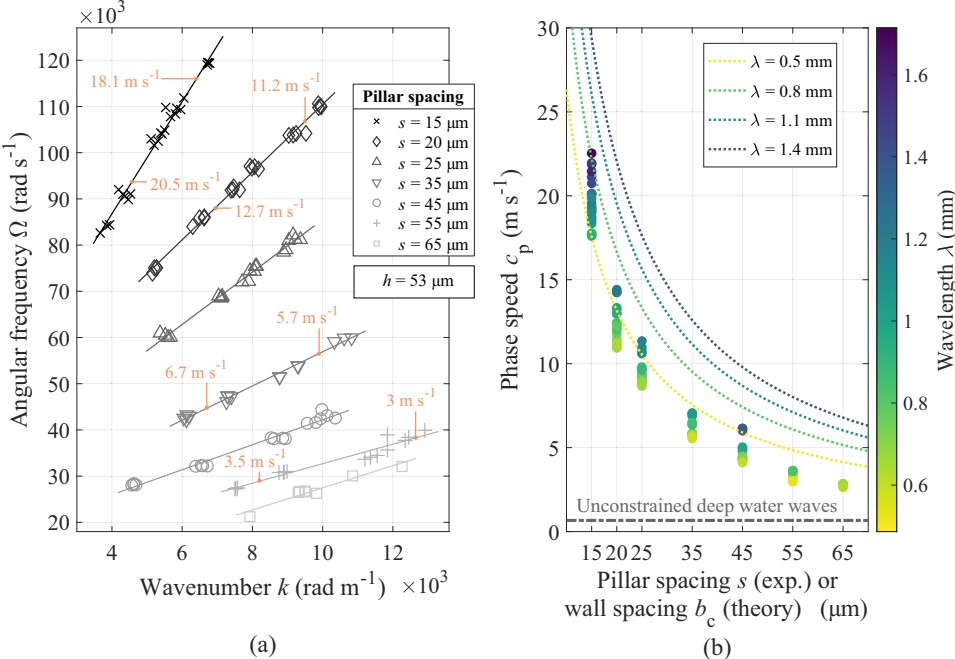

(a)                                                      (b)

**Fig. 2 | Influence of constraints and edges on the wave characteristics. a** The dispersion relation of the plastronic waves with respect to the inter-pillar spacing $s$, with fixed pillar height $h = 53\,\mu m$. Exemplary values of the ratio $\Omega/k$ (i.e., the phase speed), indicated in orange on the plot, confirm the dispersive behaviour, emphasising as the pillar spacing decreases. **b** The phase speed as a function of the pillar spacing, with fixed pillar height $h = 53\,\mu m$, all wavelengths combined. The comparison of experimental measurements is done with the theoretical phase speed of conventional (unconstrained) deep water waves ($\lambda = 1\,mm$) and with the semi-empirical model of Scott and Benjamin[12], defined in Eq. (2). This model, which describes the phase speed of an interfacial wave travelling in a deep water-filled channel as a function of the wall spacing $b_c$, plotted here for different wavelengths, shows a good agreement with the experiments. Source data are provided as a Source Data file.

plastron response to a short US pulse (50 cycles, 2.5 MHz), which is further documented in SI, Section 1.A, and illustrated in Fig. S5–7.

**Influence of constraints and edges on the wave characteristics**
With constant pillar height ($h = 53\,\mu m$), we tune the interface three-phase contact density by employing various pillar spacings $s$: 15, 20, 25, 35, 45, 55 and 65 $\mu m$. The dispersion relation of the produced plastronic waves is illustrated in Fig. 2a. All these wavenumber-to-frequency linear relations have a non-zero, positive y-intercept, meaning that dispersion occurs. This is confirmed on the graph by exemplary values of the ratio $\Omega/k$, which varies with respect to $k$. The dependence of the slope on pillar spacing suggests that the dispersion relation of the plastronic waves can be modulated by varying the pillar spacing and, thus, the density of the three-phase contact line.

The phase speed of plastronic waves grows with increasing wavelength, which is more obvious in Fig. 2b. This contradicts with the behaviour of conventional deep water capillary waves, as shown in SI, Fig. S1. Another contrasting feature of the plastronic waves is that they travel considerably faster than deep water waves, as computed from Eq. (1), for $\lambda = 1\,mm$. This represents a speed-up factor of up to 45, which seems to be facilitated by the plastron configuration. Among all our experiments, the highest phase speed achieved is 22.5 m s⁻¹, which is experienced by a 13.1 kHz plastronic wave with wavelength 1.7 mm, in the case of the smallest pillar spacing $s = 15\,\mu m$ ($h = 53\,\mu m$).

Figure 2b also depicts the semi-empirical model of Scott and Benjamin[12], which describes the phase speed $c_p$ of an interfacial wave travelling in a deep water-filled channel, as follows

$$c_p^2 = \frac{1.2\,(g + k^2\sigma/\rho) + 12\sigma/(\rho b_c^2)}{k[\coth(kh_c) + 0.0305(kb_c) - 0.000376(kb_c)^3]},\quad (2)$$

where $h_c$ and $b_c$ are the wall height and spacing. The stiffness of the interface, augmented by its pinning to the solid structures, grows with the narrowing of the space between them, and thus with the density of edge constraints[10]. Equivalently, decreasing the spacing between the micropillars will make the interface appearing stiffer from the per-spective of the propagating wave, and thus is anticipated to increase the wave speed, which is confirmed by Fig. 2b. For the largest spacing, the phase speed of the plastronic waves approaches that of conven-tional waves in deep water, as it should occur with the removal of constraints[11]. These experimental results are in line with the properties of capillary waves in water-filled channel, for which the propagation speed decreases with the broadening of the channel in which they travel[10,12]. Although the model of Scott and Benjamin focuses on centimetre-scale water waves with wave speed not exceeding 2 m s⁻¹ [12], it was here computed for a range of wavelengths ($\lambda = 0.5, 0.8, 1.1,$ and 1.4 mm) comparable with the ones of our experimental data. The parallel with the plastronic waves is striking.

**Influence of the pillar height on the wave characteristics**
With constant pillar spacing ($s = 25\,\mu m$), we now investigate the wave behaviour with respect to the pillar height $h$: 21, 25, 37, 53 or 71 $\mu m$. The dispersion relation in Fig. 3a facilitates the comparison at a given wavenumber between the plastronic waves and the conventional deep water capillary waves, again computed from Eq. (1). The angular fre-quency of plastronic waves is higher than that of conventional waves and seems to be weakly dependent on the pillar height. In Fig. 3b, for a given wavelength (i.e., a given plotting colour), the phase speed evi-dences a quadratic-like relation with downward curve and a local maximum around $h = 37\,\mu m$. Below and above this pivotal value of pillar height, a drop in phase speed is typically observed, suggesting undemonstrated mechanisms for slowing down the wave propagation.

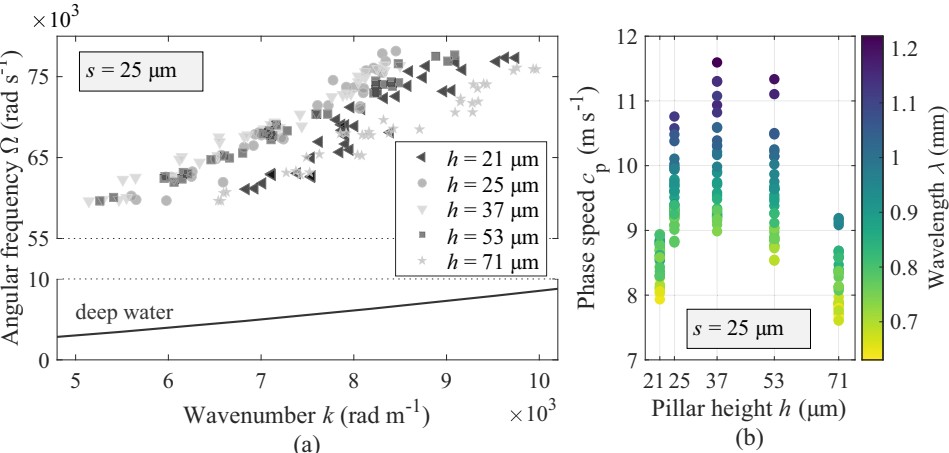

**Fig. 3 | Influence of the pillar height on the wave characteristics. a** The dispersion relation of the plastronic waves differs considerably from that of conventional deep water capillary waves with similar wavenumber. **b** The quadratic-like relation between the phase speed and the pillar height possesses a local maximum at $h = 37\,\mu m$ with decreasing speed below and above this pivotal value of pillar height. Source data are provided as a Source Data file.

In the context of ocean waves, the slowing down of shallow water waves with the thinning of the water depth is a known phenomenon. It has been repeatedly suggested by Lamb[7], Walbridge and Woodward[11] and Gjevik[38], that the phase velocity of an interfacial wave travelling on a medium's interface is weakly influenced by its viscosity, except, when the medium's depth approaches the thickness of the thin layer of viscous-dominated fluid forming close to the solid boundary. The experimental implementation of their analytical description confirmed that the motion of the water molecules driven by the swelling interface can be slowed down by this so-called viscous boundary layer, through fluid deceleration due to viscous shear stresses. An analogous force in the case of air is expected to be present, but considerably smaller, since air is about 50 times less viscous than water[39]. In the frequency range 9.5–12.1 kHz of the plastronic waves studied as a function of the pillar height, the viscous boundary layer associated to a wave perturbation in air (kinematic viscosity $v \cong 15.6 \times 10^{-6}\,m^2\,s^{-1}$, at 25 °C) has a thickness $\delta = \sqrt{v/\pi f}$[11] $\cong 21.5 \pm 1.3\,\mu m$. Because the smallest pillar heights ($h = 21$ and $25\,\mu m$) investigated in this work are such that $\delta \cong h$, the slowing down of the plastronic waves due to viscous effects as the plastron gets thinner cannot be excluded as a participating mechanism influencing the wave characteristics.

All considered, this local maximum formed around 37 μm suggests that more than one mechanism compete. Nevertheless, as the physics of the exposed wave phenomena is not predicted by the existing theory, developing a new model of interfacial wave propagation emulating these experimental conditions is necessary to thoroughly explain the behaviour of the plastronic waves.

**Plastronic waves as a monitoring tool for plastron longevity and stability**

After all, we can affirm that plastronic waves carry the properties of the plastron under investigation. Assuming the phase speed known for a given microstructured design of superhydrophobic surface, a shift of that value would be an evidence of variation in shape, volume or internal pressure of the plastron. Building on this premise, the change in phase speed of the plastronic waves has been monitored over time for a same plastron. For the investigated configuration ($h = 53\,\mu m$, $s = 25\,\mu m$), the results are shown in Fig. 4, where the mean value and the standard deviation of the phase speed are depicted as a function of time for two different values of pulse repetition period (PRP), i.e., 5 seconds in Fig. 4a and 20 seconds in Fig. 4b, and for two different conditions of gas concentration of the surrounding water. In a condition of supersaturation of gas in water, as confirmed with pH

measurements and dissolved oxygen probing (details in "Methods"), but also with the observation of spontaneous formation of bubbles on the tank walls, a relative downward shift of the phase speed is recorded over time, drifting by $9 \pm 5\%$, after 4 minutes. In a condition of undersaturation of gas in water (degassed), a relative upward shift of the phase speed is recorded over time, drifting by $8.5 \pm 4.5\%$ after 4 minutes. The variation of hydrostatic pressure $\Delta P_{hyd} = 200 \pm 10$ Pa and water temperature $T_0 = 25.7 \pm 0.4\,°C$ having not significantly changed along these sets of experiments, one can consider that the shift in phase speed reflects the plastron alteration via the gas diffusion from and to the surrounding water. When the water is supersaturated, its outgassing can spontaneously take place, e.g., via outward transfer into the plastron[24,40]. Subsequently, the plastron inflates, revealed by the observed decrease of the phase speed demonstrated in Fig. 4. In contrast, when the water is undersaturated, the water ingassing spontaneously takes place, e.g., via gas intake from the intra-plastronic gas[24,41]. Consequently, the plastron is expected to be dissolving, which is transcribed by the gradual increase of the waves phase speed observed in Fig. 4. This happens similarly with other plastron configurations, such as reported in SI for $h = 21\,\mu m$, in Fig. S8b. With these considerations, the plastronic waves become an interpreter of the plastron changes and stability. The possibility to probe a micrometre scale plastron with a remotely introduced energy in sub-millimetre length scale, i.e., the plastronic waves, can become greatly powerful, when the plastron Cassie-to-Wenzel transition and invasive methods to probe it are not viable options.

A similar shift of phase speed over time for different PRP informs us that the US actuation is not accelerating the gas diffusion, unlike a water flow would[23]. Instead, the plastronic waves could act as an effective, non-destructive probing sensor, incorporated in an active control system, for plastron regulation and wetting prevention. Considering that a key challenge in maintaining an underwater superhydrophobic surface unwet remains to this date in solutions for preventing a Cassie-to-Wenzel state transition from occurring, this successful utilisation of plastronic waves for monitoring a plastron might open up novel applications in, e.g., on-chip biofilm reactor, micromixing, leaky-wave systems, micro-/soft-robotics, implantable biosensors and microfluidic transport.

## Discussion
The marriage of surface engineering and nonlinear focused US (2.5 MHz) allowed us to reveal a novel class of interfacial waves travelling on an underwater thin gas layer in micrometre width scale. We

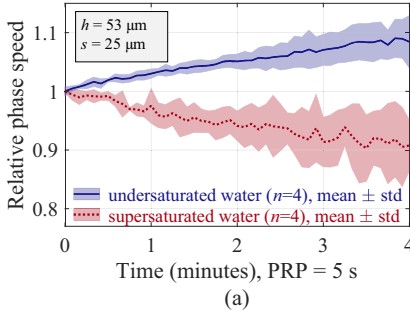
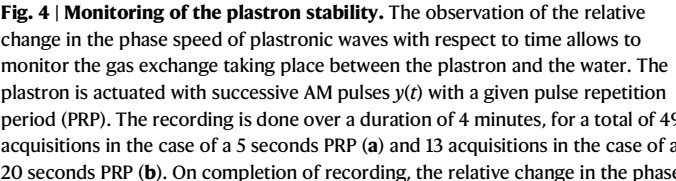
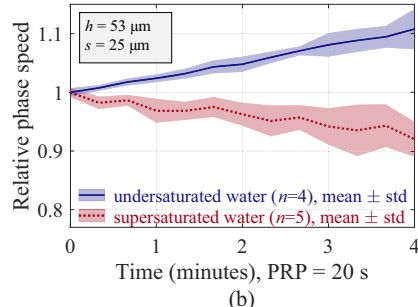

(a)                                                         (b)

**Fig. 4 | Monitoring of the plastron stability.** The observation of the relative change in the phase speed of plastronic waves with respect to time allows to monitor the gas exchange taking place between the plastron and the water. The plastron is actuated with successive AM pulses $y(t)$ with a given pulse repetition period (PRP). The recording is done over a duration of 4 minutes, for a total of 49 acquisitions in the case of a 5 seconds PRP (**a**) and 13 acquisitions in the case of a 20 seconds PRP (**b**). On completion of recording, the relative change in the phase speed significantly differentiates between the two explored configurations, in blue for a gas-undersaturated water and in red for a gas-supersaturated water, revealing the spontaneous dissolution and inflation, respectively, of the superhydrophobic plastron under monitoring. A similar behaviour observed in both cases of PRP indicates that the US actuation and the propagating waves do not accelerate the gas exchange processes. The parameter $n$ refers to the number of repetitions for each experimental configuration. Source data are provided as a Source Data file.

demonstrated that a plastron, i.e., a three-phase construct consisting of a gas layer trapped between the topography of PDMS micropillars immersed in water can, with contribution of surface tension and contact line forces, altogether form a meta-medium that can carry interfacial waves with frequency in the range 3.3–19 kHz and new characteristics, hence the name plastronic wave. High-speed optical microscopy provided means to highlight their extraordinary fast phase speed considerably exceeding that of conventional capillary waves with similar wavelength, as well as their nonlinear relation with the geometry of the microstructures and the gas layer.

To explain the behaviour of the US-induced plastronic waves, the well known complementary case of waves travelling in deep water was considered, but also the literature on waves propagating on constrained interfaces or in the vicinity of a viscous boundary layer. While the impact of increasing density of the three-phase contact line on the wave speed follows known trends[12], the quadratic-like association between the phase speed and the pillar height suggests that mechanisms not predicted by previous studies compete in the dispersion relation, at the investigated wave frequencies. Although a tentative analogy was done with the literature on gravity-capillary waves slowing down due to viscous forces[11,38], further investigations will be necessary to explain thoroughly the experimental results reported here.

Finally, the usefulness of the US-induced plastronic waves as a non-destructive interpreter of the plastron stability, via the examination of the relative change of their phase speed over time, is also demonstrated. The significance of these observations extends far and wide, appealing to a diverse audience working in applied physics, engineering, meta-materials, surface coatings, microfluidics, lab-on-a-chip and biomedical applications.

## Methods

### Microfabrication

For the microfabrication of the superhydrophobic samples, PDMS was selected for its optical transparency and compatibility with soft lithography method. Each sample consists of an array of cylindrical micropillars of constant diameter (20 μm), arranged in a square lattice, spread across an area of 5 mm × 5 mm enclosed in a 50 μm thick wall. Two series of samples differentiating by their varying parameter were fabricated. In the first series, the varying parameter was the pillar height, equalling 21, 25, 37, 53, and 71 μm, while keeping constant the pillar spacing (25 μm). In the second series, the varying parameter was the pillar spacing, equalling 15, 20, 25, 35, 45, 55 and 65 μm, while keeping constant the pillar height (53 μm). For more details, refer to SI, Section 1.C, Fig. S9 and Table S1.

### Experimental setup

As schematised in Fig. 1, a microstructured PDMS sample (manufactured as described in "Microfabrication") is immersed in pure water (Milli-Q®, 18.2 MΩ cm, 1 ppb total organic carbon), held to a manual 2-axis translation stage (Standa Ltd., 7T175-100, Lithuania) and placed at both the foci of a HIFU transducer (Sonic Concepts™, H-147, central frequency $f_{ac}$ = 2.5 MHz, Washington, USA) and a high-speed camera (Vision Research™, Phantom v1612, New Jersey, USA). The visualisation is done in a top-view perspective, through the central opening of the transducer, with a ×5 magnification objective (Canon Inc., MP-E 65 mm, Japan), resulting in an image scaling of 5.7 μm pixel⁻¹. When the sample is submerged in the water, air remains trapped between the microstructures, thanks to the interfacial tensions taking place at the three-phase boundaries, forming then a plastron. The transducer is facing the microstructured side of the sample and thus the tips of the micropillars. A controlled diving speed of <10 mm s⁻¹ assures us of a reproducible thickness of the plastron[42]. In addition to that, a constant immersion depth (5.6 ± 1.2 cm) of the sample allows to replicate the hydrostatic pressure. Once immersed, an amplitude-modulated US pulse (driving frequency 2.5 MHz, 3 cycles of AM, 0.5 V amplitude input voltage) is generated by an arbitrary function generator (B&K Precision, 4053b, California, USA) and then amplified 50× by a high voltage amplifier (Falco Systems B.V., WMA-300, Netherlands). The produced axisymmetric acoustic field was measured at the focal plane using a needle hydrophone (Precision Acoustics Ltd., 0.2 mm, England). The mechanical interaction between the US pulse and the superhydrophobic surface is recorded at high speed (159090 Hz, 384 × 176 pixels frame size, 357 ns exposure time and 80,000 exposure index), via the software Phantom Camera Control (Phantom®, version 3.4, New Jersey, USA). The synchronisation between the signal generator and the video acquisition is so that the time $\tau = 0.5$ μs (pulse generation delay, as set in the signal generator) always corresponds to the instant at which the US pulse is initiated. It should be noted that each experimental iteration is conducted with a new, fresh superhydrophobic surface, so that every experiments meet the same initial conditions.

The geometrical and acoustic properties of the HIFU transducer, as described by the manufacturer, are given in what follows. The transducer with central frequency 2.5 MHz has an outer diameter of 60 mm and an inner diameter (central opening) of 22.6 mm. The curvature radius at radiating surface is 50 mm, and the focal depth is 39 mm. The pressure focal gain is 91.17, assuming 1 at the radiating surface and in a linear homogeneous field. The focal width and length at half-amplitude (−6 dB) equal 0.51 mm and 3.28 mm.

## Tuning of gas concentration of water

In the run of experiments evaluating the impact of gas concentration on the rate of gas diffusion from and to the plastron, via the continuous analysis of US-induced plastronic waves, the Milli-Q water was degassed with a vacuum system (Vacuum Chambers, 2RS-3, Poland) and then carbonated by the means of a sparkling water device (SodaStream International Ltd., Jet™, Israel). Calibrated pH metre (Mettler Toledo™, Seven Compact, Switzerland) and dissolved oxygen metre (Milwaukee Instruments Inc., MW600, North Carolina, USA) were employed to measure the concentration $C_o$ of carbon dioxide and the concentration $D_o$ of dissolved oxygen present in the Milli-Q water throughout the experimental runs. Based on the pH measurement, the concentration $C_o$ is calculated from the expression $C_o = 10^{-2 \cdot pH}/K_{a1}$, where $K_{a1}$ is the first constant for dissociation of carbonic acid $H_2CO_3$ in water[43]. Our gas-supersatured water had a $CO_2$ concentration ranging between $C_o = 153 \pm 89\,\mu M$ (pH $= 5.09 \pm 0.1$) and $C_o = 304 \pm 178\,\mu M$ (pH $= 4.94 \pm 0.1$), and a concentration $D_o$ in the range $3.1$–$3.2 \pm 0.1\,mg\,L^{-1}$. For the undersaturation experiment, the water was placed for 15 minutes in the vacuum system for depressurisation ($-0.98 \pm 0.01$ bar). After degassing, the $CO_2$ concentration ranged between $C_o = 7.3 \pm 8\,\mu M$ (pH $= 5.75 \pm 0.1$) and $C_o = 15.3 \pm 9\,\mu M$ (pH $= 5.59 \pm 0.1$). The concentration $D_o$ was in the range $2.8$–$3.2 \pm 0.1\,mg\,L^{-1}$. If the gas concentrations happened to be significantly below or above these concentrations, the plastron was quickly (<4 minutes) experiencing, respectively, collapse or depinning. An example of such depinning event is depicted in SI Fig. S8a. As the intra-plastronic gas volume and pressure increase, so does the Laplace pressure. When the interfacial tensions cannot hold the bulging interface, the gas-water interface detaches (i.e., depinning) from the micropillar tips and a tethered bubble forms, as presented in more detail in a recent work[44].

While $D_o$ was on the same order of magnitude for both investigated configurations of gas concentration, $C_o$ significantly differed. This confirms that the concentration of carbon dioxide in the water, as investigated here, was the parameter driving the spontaneous ingassing and outgassing of the plastron. While the results presented in Fig. 4 were produced with $h = 53\,\mu m$, another set of experiments conducted with $h = 21\,\mu m$ and with the same experimental conditions led to similar results, as illustrated in SI Fig. S8b. More details are provided in SI, Section 1.D.

## Image analysis

The phase speed $c_p$, wavelength $\lambda$, frequency $f$ and attenuation ratio $\alpha$ of the US-generated waves result from the analysis of the wavefronts, i.e., the wave lines of same phase, performed in a polar coordinate system, with an home-made Matlab routine. A brief description of the method is given here, and supplemented in SI by details in Figs. S10 and S11, commented in Section 1.E.

The conversion of the high-speed video information into a polar coordinate system, the origin being the image-detected centre of the acoustic actuation, allowed to display the waves into a time $vs.$ radial distance arrangement, as exemplified in Fig. S10. After a detection of the wavefronts, the wave speed $c_p$ is evaluated, based on the slope. Then, the wave period $T$ is obtained by computing the average time spacing between wavefronts with same phase, as shown in Fig. S11. Eventually, the wavelength $\lambda$ and the wave frequency $f$ are such that $f = 1/T$ and $\lambda = c_p/f$.

## Data availability

The raw data generated in this study have been deposited in the Zenodo database under accession code https://doi.org/10.5281/zenodo.14393997. An exemplary script (Supplementary Data 1) and its associated video file (Supplementary Data 2) for wavefronts analysis are provided as Supplementary Information. Source data are provided with this paper.

## Code availability

Additional custom algorithms are available upon request.

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

## Acknowledgements

The authors thank the Research Council of Finland for funding this research via grant agreements No. 342169 (R.H.A.R.) and 342170 (H.J.N.), and via the Centre of Excellence Programme 2022-2029, in Life-Inspired Hybrid Materials (LIBER), project number 346109 (R.H.A.R.). W.S.Y.W. acknowledges the funding support from the Research Council of Finland as well, grant agreement No. 13347247, and also from the European Union's HORIZON research and innovation program under Marie Skłodowska-Curie grant agreement No. 101062409. M.F. acknowledges the Finnish Cultural Foundation (grant No. 240398). All colleagues from the Medical Ultrasonics Laboratory and the Soft Matter and Wetting group are warmly thanked for their constructive criticism and suggestions. Professor Anton Kuzyk and his team are acknowledged for the provision of their Milli-Q water system and for the access to their pH measuring system. Professor Matilda Backholm is acknowledged for her support through the provision of the carbonation system and the guidance with $CO_2$ measurement.

## Author contributions

B.K. and W.S.Y.W. designed and fabricated the superhydrophobic samples. M.F. set up and conducted the experiments. M.F. produced and analysed the raw data. All authors, M.F., B.K., A.D.G., W.S.Y.W., R.H.A.R. and H.J.N., participated in the interpretation of the results. M.F. wrote the manuscript. All authors revised the manuscript and approved the submitted version. H.J.N. and R.H.A.R. planned and supervised the project.

## Competing interests

The authors declare no competing interests.
