## [Transparent Peer Review file · Nature Communications]

Fast capillary waves on an underwater superhydrophobic surface

Corresponding Author: Dr Maxime Fauconnier

Version 0:

Reviewer comments:

Reviewer #2

(Remarks to the Author)
Please see attached file

(Remarks on code availability)

Reviewer #3

(Remarks to the Author)

The paper focuses on the capillary waves generated at the surface of a plastron, showing some interesting results. The experiments are well-conducted, particularly considering their difficulty, the studied waves are highly dissipated and last only a few wavelengths. The influence of the microstructure is studied, showing a significant impact on the propagation of capillary waves due to (i) the interface confinement between pillars and (ii) the air confinement in the plastron. Although the experiments are well-described and analyzed, the analysis could benefit from some clarification in the main manuscript. However, the interpretation is somewhat confusing and requires a more detailed analysis to be publishable. Finally, the last part presents a potential application to monitor the plastron dynamics but seems a bit irrelevant due to the complexity of interpreting the results.

In conclusion, even though the experiments are interesting and well-performed I cannot recommend publication because the interpretation and some analysis parts present several issues as listed below :

- The authors interpret the change in phase velocity using either the influence of the wall (spacing dependency) or of the dissipation/compressibility (height dependency). However, they overlook the influence of the wavelength on the velocity, while capillary waves are dispersive. The influence of the wavelength should be taken into account or at least discussed in the paper.
- Compressibility of the air layer is invoked to explain the slowing down of the wave with height. However, capillary waves conserve volume. One could argue that local compression might occur, but as the speed of sound is 30 times faster than the observed waves, air compressibility should not play a significant role.
- The slowing of the phase velocity at small pillar height is interpreted by viscous dissipation, which seems feasible. However, the reference 42 used by the authors to support their claim is a conference paper that appears to study longitudinal waves in soft solid. The authors should look into the literature on viscous gravito-capillary waves. Moreover, the authors did not discuss other known sources of dissipation such as surface contamination, which could strongly affect the results over time, or meniscus dissipation due to flow in the vicinity of the pillars, which could depend on the thickness as well.
- Figures 4a) and b) are unreadable. The authors should plot wavelength and frequency as functions of spacing. I understand the original idea; however, the influence of pillar height is flattened by the variation in pillar spacing, as shown in figures c) and d). Moreover, if I understand correctly, the standard deviation is based on three experiments, which seems

low.

- It is not clear how to interpret the change in contrast. The authors seem to indicate that brighter means troughs and darker translates to crests. This vision seems in opposition with a "lens" effect of the interface. This part should have been more detailed. Additionally, when looking at the supplementary data the focal area becomes darker at the beginning (image 7 and 8) when the surface is expected to be pushed toward the air side by radiation pressure which also appears in contradiction with the interpretation of the author.

- No details are given on the acoustic calibration, although the acoustic field is shown in Fig. 3 and some acoustic properties are given in Table 1. More importantly, the intensity given by the authors results in a radiation pressure of around 9000 Pa ($p_{\text{rad}} \sim 2I/c$) for the lowest excitation, which is several times higher than the critical breakdown pressure estimated around 300 Pa or lower (using the sliding model with $\theta_a=120^\circ$ for the lowest spacing), and the Wenzel state should have been forced as detailed in Ref. 32. The authors should have commented on this discrepancy...

(Remarks on code availability)

Version 1:

Reviewer comments:

Reviewer #2

(Remarks to the Author)
Please see the attached file

(Remarks on code availability)

Reviewer #3

(Remarks to the Author)
The authors have made significant improvements by conducting new, better-calibrated experiments and enhancing the analysis with dispersion relation plots, which greatly strengthen the manuscript.

I can now recommend publication, provided the comments listed below are addressed.

The AM modulated excitation is effective only within a certain frequency range. The authors speculate that this is due to a "resonant-like behavior" of the plastron. However, I believe it is related to the match between the spatial forcing (HIFU field) and the generated wavelength. The HIFU imposes a wavelength of approximately 0.8 mm, corresponding to a wavenumber of 7850 m^{-1} , which is the typical wavenumber excited in the study.

Please remove the gas elasticity hypothesis. As both reviewers have already explained, this is highly unlikely since the plastronic waves are slower than the speed of sound in air, meaning air compressibility cannot play a role. While the gas flow might indeed depend on the air thickness, as suggested by the authors, it will be governed by hydrodynamic laws without significant density change.

The colors in Figures 2c and 3b are inverted; please harmonize them.

The paragraph discussing the speed of plastronic waves at the beginning of page 11 is somewhat out of context. It is not a race. Additionally, some sentences are repeated in the conclusion. I suggest removing this paragraph.

Although radiation forces are a mean effect, they act on a timescale of a few acoustic periods. The fact that the transition is not observed likely has more to do with the time it takes for the interface to move within the microstructure, which should be limited by liquid inertia, air flow, or dissipation at the contact line.

I didn't notice this in the first round, but "plastronic wave" is catchy—I like it!

(Remarks on code availability)

Version 2:

Reviewer comments:

Reviewer #2

(Remarks to the Author)
See the attached file

(Remarks on code availability)

Reviewer #3

(Remarks to the Author)
I was initially inclined to recommend publication and the new version confort this.
I have only one minor comment. I believe n in figure 4 is not defined (number of repetition ?).

(Remarks on code availability)

The manuscript describes measurements of capillary waves in a plastron, i.e. a thin layer of air supported by a hydrophobic micropillar array immersed in water. The main finding is that the capillary wave phase velocity in a plastron is much greater (up to a factor of 20) than that of conventional capillary waves at the water/air interface. The experiment is not perfect: the dispersion of capillary waves, i.e. the frequency vs. wave vector dependence, is not reported due to the lack of control over the frequency. (Perhaps one could control the capillary wave frequency by using an acoustic wave with a sinusoidal amplitude modulation instead of single tone-burst?) Another weakness is the dependence of the results on the acoustic intensity, which indicates the presence of nonlinear effects. (This should be acknowledged in the manuscript.) Unless the study is specifically aimed at studying those, it would be preferable to conduct it in the linear regime. However, a plastron is an interesting system for studying capillary waves, and I believe the originality of the experiment outweighs its weaknesses. This experiment is likely to be of interest to a broad audience of scientists beyond those studying water waves. However, the manuscript contains multiple deficiencies, primarily in the interpretation and presentation of the results, that make it unsuitable for publication in its current form.

1. I believe that the notion of “shallow gas” is misguided. Due to the very low density of air, the second term in the denominator of Eq. (1) is negligible for the wavelength range $650 - 800 \mu\text{m}$ and layer thicknesses $20 - 70 \mu\text{m}$. Consequently, even a $20 \mu\text{m}$ thick plastron should be considered infinitely thick for the purpose of the capillary wave propagation. From Fig. 5(a), it is obvious that the large phase velocity is caused by the high density of micropillars rather than by the small thickness of the plastron. While Fig. 5(b) does show a moderate dependence of the phase velocity on the plastron thickness, it is inconclusive, as the data shown in the figure correspond to neither constant frequency nor constant wave vector.
2. The authors are correct in noting an analogy with the propagation of capillary waves in narrow channels described in Ref. 11. However, they seem to overlook the fact that Ref. 11 deals with the situation when the contact line is pinned to the edge of the channel. It is the pinning effect that causes the increase of the phase velocity: as demonstrated experimentally in Ref. 9, in channels with slipping contact line conditions, the phase velocity is the same as on an unconstrained water surface. Likewise, in the present experiment, it is likely that the pinning of the interface by micropillars makes the interface effectively stiffer and increases the capillary wave phase velocity. I’m afraid this has very little to do with “general relationships of mechanical waves and medium’s mechanical impedance” mentioned on p. 8 of the manuscript.
3. Fig. 5(a) presents the most important result, it would be nice to highlight it more. It would be helpful to indicate the conventional deep water capillary wave velocity on this figure – that will make Fig. 1 superfluous. The authors state that the measured phase velocity shown in Fig. 5(a) “appears to asymptotically converge towards the theoretically predicted phase speed of conventional unconstrained capillary waves.” I do not see any evidence of such asymptotic convergence in the data, although it is of course expected in the limit of infinite pillar spacing. (However, the plastron will no longer exist at that limit.) Please specify which equation from Ref. 11 was used to calculate the red curve.
4. The information presented in other panels in Figs. 4 and 5 is somewhat poorly organized. The way the data are presented in Fig. 4(a,b) is especially confusing. Why not show conventional $\gamma(x)$ plots, e.g. wavelength as a function of pillar spacing at constant height and wavelength as a function of pillar height at constant spacing? Adding the acoustic intensity as another parameter adds

confusion because of possible nonlinear effects. Why not limit the presentation to low intensity data? (High intensity data could be then shown in the supplement.) Fig. 5(c) is not too useful. The phase velocity of capillary waves is a local property and should not depend on the size of the plastron as long as the latter is much larger than the wavelength. The authors state that the velocity is lower for the larger plastron. I do not see any convincing evidence of that: what I see is that for the smaller plastron the spread in the data is larger, indicating a larger statistical error in the measurements.

5. Throughout the paper, the analysis is presented in the form of speculative verbal statements not supported by any model or quantitative estimates. While speculative statements are sometimes admissible, in a physics paper they should be used very sparingly. Some of these statements are clearly incorrect: for example, it is well known that the capillary wave propagation is not affected by the elasticity of the liquid or gas as long as the capillary wave velocity is much smaller than the speed of sound. The suggestion that the viscous boundary layer affects the capillary wave velocity appears to be incorrect either. It is clear that a sound physical model describing capillary waves in a plastron is currently lacking and I would recommend to acknowledge that it is lacking instead of offering unsupported explanations.
6. The title should be descriptive. I would recommend to mention capillary waves and plastron in the title.
7. The expressions such as “very first and unique” are inappropriate in a scientific paper, especially in the abstract.
8. In the caption to Fig. 3, please first describe panel (a), then panel (b). Please label the vertical axis.
9. The caption to Fig. S4 should describe every panel in the figure.
10. While the procedure of measuring the phase velocity is described in some details, it would be helpful to describe the procedure used to measure the wavelength and frequency.

I greatly appreciate a new set of experiments conducted with an improved methodology. The presentation has also been improved. I would like to reiterate that this is an interesting study that will be of interest to a wide audience. However, I still find that the notion of “shallow gas” is misguided and not supported by the experimental results presented in the manuscript. The authors show very clearly that the main result, i.e., the high phase velocity of capillary waves in the plastron, originates from constraining the motion of the interface by the pillars. This has nothing to do with the “shallowness” of the plastron. There is indeed a weak dependence on the pillar height, therefore I agree that on the basis of the experimental data, the effect of the finite plastron thickness cannot be excluded. However, the data pertaining to the dependence on the pillar height are not nearly as clear-cut as the dependence on the pillar spacing. For example, the authors contend that “a systematic drop in phase speed is observed for the pillar heights (21 and 25 μm) matching the thickness of the viscous boundary layer inside the plastron.” However, according to Fig. 2(b), at some wave vectors the structure with $h = 25 \mu\text{m}$ yields the highest phase velocity. Furthermore, the authors suggest that capillary waves may get slowed down by viscosity of the “shallow” air layer, but the main effect they observe is a big increase in the phase velocity. The cited references pertaining to the viscosity effect (with the exception of Ref. [42], which is entirely irrelevant as it deals with acoustic waves rather than water waves) consider the viscosity of the liquid. The expectation that the air viscosity will cause an analogous effect is unfounded: whatever happens in the air has very little effect on the capillary wave propagation because of the very low density of air. The authors admit that the dependence of the phase velocity on the plastron thickness is not well understood and further research is needed. I find that the emphasis on the “shallowness” of the air layer in the abstract, introduction, and conclusion is not warranted and may even be misleading, as it may lead the reader to believe that the reported high phase velocity results from the shallow air layer.

There are several other issues that need to be addressed:

1. The abstract claims that “a nonlinear relation of the propagation speed with the gas layer geometry” is an “unprecedented feature” of plastronic capillary waves. I don’t think this claim is well founded. The phase velocity of conventional capillary waves is a nonlinear function of the depth of the liquid according to Eq. (1). I would also recommend to avoid superlatives such as “unprecedented.” It should be left to the reader to decide whether the results presented in the paper are unprecedented or not.
2. The data for $s = 25 \mu\text{m}$, $h = 53 \mu\text{m}$ shown in Fig. 2(b) don’t seem to match the data for the same plastron in Fig. 3(a).
3. What does the “shallow water” curve in Fig. 2(b) show? The phase velocity of capillary waves on shallow water depends on the water depth. I would recommend removing this curve.
4. I cannot quite make sense of the following text: “The adhesive forces holding the plastron follow a quadratic growth with increasing density of three-phase contact line [44]. As a result, this will make the interface appearing stiffer from the perspective of the propagating wave.” Why should the adhesive force between the droplet and the plastron discussed in Ref. [44] affect the capillary wave propagation? I would think that it’s the pinning of the interface by the pillars that makes it effectively stiffer.

A few minor points:

5. It is obvious that the capillary wave frequency should be equal to twice the amplitude modulation frequency. Its not necessary to show 5 graphs demonstrating this in Fig. 2, one set of data would suffice.
6. The last two sentences in the 2nd paragraph on p. 6 are repeated in the caption to Fig. 1. This is unnecessary.

Reviewer #2 (Remarks to the Author):

7. On p. 13, frequencies from 3.3 to 19 kHz are referred to as “low frequencies”. For capillary waves, these frequencies aren’t low.
8. I would replace the word “eventually” in the beginning of the last paragraph of Conclusions by “finally.”

I appreciate the revisions made by the authors. The manuscript has been sufficiently improved to recommend publication. I have a few optional recommendations that the authors may want to consider.

1. In Fig. 2(a), it looks as if the beam from the light source passes through the HIFU transducer. It would be helpful to clarify this point.
2. What is SHS in Fig. 2(a)?
3. It would be helpful to present a single-frame image (perhaps in the SI) showing circular capillary waves.
4. P. 2: "Interfacial waves on liquids are generally categorised, according to their wavelength into two distinct groups, capillary ($\lambda \leq 1.7$ cm) and gravity ($\lambda \geq 1.7$ cm) waves [...]"
Since the length 1.7 cm is specific to pure water and will be different for other liquids, consider replacing "liquids" by "water".
5. P. 4: "This essentially results from the acoustic waves having sufficient time-averaged energy density to provide an ARF overcoming the resisting forces arising from the local interfacial tensions." This sentence can be incorrectly interpreted to imply that the ultrasonic power should overcome a certain threshold in order to excite capillary waves. Please consider revising.
6. P. 7: "side-view demonstration of plastron deformation performed in a superhydrophobic micro-channel." Do I understand it correctly that the microchannel shown in Fig. S3 is not a plastron? Then the expression "demonstration of plastron deformation" is inaccurate.
7. Placing equations in figures as in Fig. 2(b) is somewhat unusual. It may be more appropriate to present this equation in the text.
8. P. 8: "depletion of constraints." I would recommend to replace "depletion" by "removal".
9. P. 8: "the model of Scot and Benjamin focuses on millimeter-scale water waves." Did the authors mean to say "centimeter-scale"?
10. P. 12: "further investigations will be necessary to understand the exact mechanisms driving the plastronic waves." I believe in the experiment described in the manuscript the plastronic waves are driven by the acoustic radiation pressure and this is well understood? What is not understood is the dependence of the phase velocity on the pillar height.

Authors' Response to Reviews of

Fast capillary waves on an underwater superhydrophobic surface

Maxime Fauconnier, Bhuvaneshwari Karunakaran, Alex Drago-González, William S. Y. Wong, Robin H. A. Ras and Heikki J. Nieminen.

Nature Communications, Research Article, NCOMMS-24-02897-T

RC: Reviewers' Comment, AR: Authors' Response and text corrections

AR: We greatly appreciate the thorough and insightful feedback provided by the Reviewers. The comments and suggestions have helped us to significantly improve the manuscript. We have carefully revised the manuscript according to the suggestions provided by the Reviewers.

Below, we address each of the Reviewers' comments point-by-point.

Reviewer #2 (Remarks to the Author):

RC: *The manuscript describes measurements of capillary waves in a plastron, i.e. a thin layer of air supported by a hydrophobic micropillar array immersed in water. The main finding is that the capillary wave phase velocity in a plastron is much greater (up to a factor of 20) than that of conventional capillary waves at the water/air interface. The experiment is not perfect: the dispersion of capillary waves, i.e. the frequency vs. wave vector dependence, is not reported due to the lack of control over the frequency. (Perhaps one could control the capillary wave frequency by using an acoustic wave with a sinusoidal amplitude modulation instead of single tone-burst?) Another weakness is the dependence of the results on the acoustic intensity, which indicates the presence of nonlinear effect. (This should be acknowledged in the manuscript.) Unless the study is specifically aimed at studying those, it would be preferable to conduct it in the linear regime. However, a plastron is an interesting system for studying capillary waves, and I believe the originality of the experiment outweighs its weaknesses. This experiment is likely to be of interest to a broad audience of scientists beyond those studying water waves. However, the manuscript contains multiple deficiencies, primarily in the interpretation and presentation of the results, that make it unsuitable for publication in its current form.*

AR: We thank the Reviewer for the positive evaluation.

Assessment of the phenomena for sinusoidal actuation at controlled frequencies has now been provided, as suggested by the Reviewer. This is elaborated more thoroughly within the response letter below.

Comment 1

RC: *I believe that the notion of “shallow gas” is misguided. Due to the very low density of air, the second term in the denominator of Eq. (1) is negligible for the wavelength range 650 – 800 μm and layer thicknesses 20 – 70 μm . Consequently, even a 20 μm thick plastron should be considered infinitely thick for the purpose of the capillary wave propagation. From Fig. 5(a), it is obvious that the large phase velocity is caused by the high density of micropillars rather than by the small thickness of the plastron. While Fig. 5(b) does show a moderate dependence of the phase velocity on the plastron thickness, it is inconclusive, as the data shown in the figure correspond to neither constant frequency nor constant wave vector.*

AR: We agree with the Reviewer. To address this issue, a completely new series of experiments has been conducted. Originally, a short ultrasound pulse (50 cycles, 2.5 MHz, not modulated) with constant amplitude was used. The new series of experimental employs instead an amplitude-modulated (AM) pulse as the driving force, as suggested by the Reviewer. By doing so, the plastronic waves were forced to oscillate at a single frequency, which were confirmed to be twice the AM frequency, as illustrated in the following Fig. 2(a) and in the manuscript through the following lines:

With constant pillar spacing ($s = 25 \mu\text{m}$), we tune the plastron thickness h_g by employing various pillar heights h : 21, 25, 37, 53 or 71 μm . In Fig. 2(a), the frequency of the plastronic waves is systematically studied as a function of the signal AM frequency. When the latter varies in the range 4.8-6.3 kHz, the frequency of the plastronic waves varies in the range 9.5-12.1 kHz. A ratio 2:1 of the wave frequency to the AM frequency is thus repeatedly observed, regardless of the pillar height, confirming that the AM acoustic radiation force poking the plastron with periodicity $T_{AM}/2$ is the driving mechanism of the plastronic waves. The determination of the range of AM frequencies employed for each microstructure geometry, which is documented in SI, Section 1.A, and illustrated in Fig. S4, results from experimentation on the plastron response to an US short pulse (50 cycles, 2.5 MHz). Further information on the characteristics of the produced waves is given in Fig. S7 and S8.

Figure 2: (a) The wave frequency as a function of the AM frequency for different pillar heights ($s = 25 \mu\text{m}$) always exhibits a 2:1 ratio, demonstrating the successful control on the frequency of the produced plastronic waves.

Therefore, this new experimental method allowed to study the phenomena at one frequency at a time, as suggested by the Reviewer, facilitating results, analysis and interpretation. Because the results of the original manuscript (produced with the short US pulse) are still of interest, we have moved some of this data from the original manuscript to the Supplementary Information (SI), as in Fig. S4, S7 and S8, replicated below.

Figure S4: The natural frequency of the plastronic waves generated by a single US pulse (50 cycles, 2.5 MHz, no amplitude modulation) as a function of (a) the pillar spacing s (with constant $h = 53 \mu\text{m}$) or (b) the pillar height h (with constant $s = 25 \mu\text{m}$). These natural frequencies were used to define the central frequency of the range of amplitude-modulated frequencies employed to drive the plastron of a given configuration of micropillars.

Also, to minimize possible nonlinear effects due to using different acoustic intensities across all data, this new series of experiments was conducted at a constant driving amplitude, *i.e.*, 0.5 V driving voltage amplified 50 times, which corresponds to *ca.* 1.5 MPa focal acoustic pressure, according to free-field hydrophone measurements. The new results still support both claims, *i.e.*, (i) the influence of pillar spacing (driving the interfacial tension) and (ii) a second mechanism associated with pillar height. Therefore, the main conclusions remain globally unchanged compared to the original version of the manuscript.

Figure S7: Wavelength-frequency analysis, resulting from the tracking and analysis of capillary waves propagating on the plastron of a SHS. The average value and standard deviation of their wavelength (a) and frequency (b) are shown as a function of the pillar height and the pillar spacing, or solid-liquid contact area fraction. A detail of the wavelength (c) and frequency (d) as a function of the pillar height (pillar spacing = 25 μm) and the driving pulse-average acoustic intensity is provided. The latter is expressed in mW cm^{-2} .

Figure S8: Phase speed as a function of the pillar spacing (a), the pillar height (b) and the plastron volume (c), respectively, with fixed pillar height ($53 \mu\text{m}$), fixed pillar spacing ($25 \mu\text{m}$) and fixed pillar height ($53 \mu\text{m}$) and spacing ($25 \mu\text{m}$). For what concerns (a), our experimental results are compared with the semi-empirical law of Benjamin and Scott [3] drawn in solid red line, describing the phase speed of an interfacial wave ($\lambda = 800 \mu\text{m}$) travelling in a deep water-filled channel. To subfigures (b) and (c), the driving force expressed in mW cm^{-2} is informed. The attenuation ratio α of the same set of recorded waves is depicted in (d), all driving amplitudes combined.

Figure 2: (b) The dispersion relation of the plastronic waves importantly differentiates them from conventional deep and shallow water waves, demonstrating the unique behavior of the plastronic waves. (c) The quadratic-like relation between the phase speed and the pillar height possesses a local maximum at $h = 37 \mu\text{m}$ with decreasing speed below and above this pivotal value of pillar height.

It is true that the computation of Eq. (1) for the case of an air layer thickness $h = 20 \mu\text{m}$ gives a similar result as for an infinitely thick air layer ($h \gg \lambda$). As a reminder, Eq. (1) is the following,

$$\Omega^2(k) = \frac{(gk + \frac{\sigma}{\rho_a} k^3) (\rho_a - \rho_b)}{\rho_a \coth(kh_a) + \rho_b \coth(kh_b)}, \quad (1)$$

However, literature generally refers to shallow water waves as interfacial waves being affected by the bottom of the water phase, and not only as interfacial waves on a medium layer, such that $h \ll \lambda$. The theoretical model expressed by Eq. (1) does not consider the viscosity of the investigated media, and thus neglects the possible boundary layer effects. $h \ll \lambda$ is not a sufficient condition for defining a shallow medium, given that the condition $h \simeq \delta$, where δ is the thickness of the viscous boundary layer, can also have a major effect on the waves behaviour, as suggested by Lamb [9], Walbridge and Woodward [13] and Gjevik [6]. A similar effect was reported by Ward et al. [14] in the context of acoustic waves travelling through a narrow slit, in air, and slowed down by to the presence of a viscous boundary layer occupying 5 % of the slit width. In our experimental configuration, $h \simeq \delta$ at the investigated frequencies. Therefore, we believe that the plastronic waves configuration possesses some characteristics of waves in shallow media, *i.e.*, interfacial waves affected by the proximity of a rigid wall. While it is not methodically demonstrated that this governs the phase speed of plastronic waves at small pillar heights, as suggested in Fig. 2c, this cannot be excluded as an influencing mechanism.

As a conclusion, we agree with the Reviewer that the speculations expressed so far in the manuscript must be trimmed. The discussion on the influence of the pillar height on the phase speed was modified with the

following comments, now added to the section 2.3:

For a given wavelength, the results of Fig. 2(c) evidence a quadratic-like relation of the phase speed and the pillar height with downward curve and a local maximum around $h = 37 \mu\text{m}$. Below and above that value of pillar height, a drop of phase speed is systematically observed, highlighting the action of a previously undemonstrated mechanisms slowing down the wave propagation.

In the context of ocean waves, the slowing down of shallow water waves with the thinning of the water depth is a known phenomenon. It has been repeatedly suggested by Lamb [9], Walbridge and Woodward [13] and Gjevik [6] that the phase velocity of an interfacial wave travelling on a medium's interface is weakly influenced by its viscosity, except, when the medium's depth approaches the thickness of the thin layer of viscous-dominated fluid forming close the solid boundary. The experimental implementation of their analytical description confirmed that the motion of the water particles driven by the swelling interface can be slowed down by this so-called viscous boundary layer, through fluid deceleration due to viscous shear stresses. A similar mechanism in the context of acoustics in narrow slit cavities was also reported more recently and likewise attributed to boundary-layer drag forces [14].

An analogous force in the case of air is expected to be present, but considerably smaller, since air is about 50 times less viscous than water [1]. In the frequency range 9.5-12.1 kHz of the plastronic waves studied as a function of the pillar height, the viscous boundary layer associated to a wave perturbation in air (kinematic viscosity $\nu \simeq 15.6 \times 10^{-6} \text{ m}^2 \text{ s}^{-1}$, at 25°C) has a thickness $\delta = \sqrt{\nu/\pi f}$ [13], approximating $21.5 \pm 1.3 \mu\text{m}$. Because the smallest pillar heights ($h = 21$ and $25 \mu\text{m}$) investigated in this work are such that $\delta \simeq h$, the slowing down of the plastronic waves due to viscous effects as the plastron gets thinner cannot be excluded. For higher pillars ($> 37 \mu\text{m}$), a drop in phase speed is also observed, as the pillar height varies from 37 to 71 μm . The local maximum formed around 37 μm suggests that more than one mechanism compete in the dispersion relation. The displacements of the gas-water interface induced by focused US in superhydrophobic samples can be considerable, up to 20 % of the plastron thickness, as suggested by the experimentation in superhydrophobic channels presented in SI, Fig. S3. Therefore, gas flow within the plastron and the role of the viscous boundary layer and of the gas elasticity cannot be ruled out, as possible driving mechanisms. Nevertheless, as the physics of the exposed wave phenomena is not predicted by the existing theory, developing a new model of interfacial wave propagation in viscous shallow gas is necessary to thoroughly explain the plastronic waves behaviour.

... and to the Conclusion:

To explain the behaviour of the US-induced plastronic waves, well known complementary cases of waves travelling in deep and in shallow water were considered, but also the literature on waves propagating on constrained interfaces or in the vicinity of a viscous boundary layer. While the impact of increasing density of the three-phase contact line on the wave speed follows known trends [11], the influence of the gas phase thickness was not predicted by previous studies. The phase speed has a quadratic-like association with pillar height suggesting that more than one mechanism competes in the dispersion relation, at the investigated wave frequencies. Interestingly, a systematic drop in phase speed is observed for the pillar heights (21 and 25 μm) matching the thickness of the viscous boundary layer inside the plastron. Although a tentative analogy was done with the literature on gravity-capillary waves slowing down due to viscous forces [13, 6], further investigations will be necessary to understand the exact mechanisms driving the plastronic waves.

Comment 2

RC: *The authors are correct in noting an analogy with the propagation of capillary waves in narrow channels described in Ref. 11. However, they seem to overlook the fact that Ref. 11 deals with the situation when the*

contact line is pinned to the edge of the channel. It is the pinning effect that causes the increase of the phase velocity: as demonstrated experimentally in Ref. 9, in channels with slipping contact line conditions, the phase velocity is the same as on an unconstrained water surface. Likewise, in the present experiment, it is likely that the pinning of the interface by micropillars makes the interface effectively stiffer and increases the capillary wave phase velocity. I'm afraid this has very little to do with "general relationships of mechanical waves and medium's mechanical impedance" mentioned on p. 8 of the manuscript.

AR: Our original phrasing quoted by the Reviewer was indeed poorly expressed. We share thus the Reviewer's opinion and are aware that the condition of the contact line whether it is pinned to the channel edge or freely moving can significantly modify the interfacial tension, as it was already stated in the Introduction of the initial version of the present work, via the following lines:

The pinning contact of the water interface to the walls of a channel in which a wave travels can also exert an additional restoring force, opposite to the interface displacement, which further stiffens the gas-water interface [10] and speeds up the wave propagation [13, 10]. This effect is intensified, when the wall spacing is narrowed [11] and the water interface bulges out [12]. In contrary, a condition of freely moving contact line agrees with the theory of waves on constraint-free interfaces [10, 7].

Accordingly, the literature and the Reviewer's perception support that the pinning of the interface by the micropillars is the main reason for the stiffening of the plastron interface and the increase of the phase speed, as now more clearly stated in the main document. With increasing pillar spacing, these interfacial forces induced by the pinning are expected to decrease, as suggested by Dubov et al. [5], and so would the interface stiffness. As a result, the phase speed should decrease, as demonstrated by our experimental results in the revised Fig. 3(b).

Emphasis and accuracy has been placed on this aspect and addressed in the revised manuscript, through the following lines:

The adhesive forces holding the plastron follow a quadratic growth with increasing density of three-phase contact line [5]. As a result, this will make the interface appearing stiffer from the perspective of the propagating wave, and thus will increase the wave speed, which is confirmed by Fig. 3(b). Inversely, the decrease of the interfacial tensions due to a larger spacing between the solid structures (i.e., the pillars or the walls) slows down the waves. For the largest spacing, the phase speed of the plastronic waves approaches that of conventional waves in deep water, as it is expected with the depletion of edges and constraints [13]. Importantly, these experimental results are in line with the properties of capillary waves in water-filled channel, for which the propagation speed is decreasing with the broadening of the channel in which they travel [11]. Although the focus of their model is on millimeter-scale water waves with wave speed not exceeding 2 m s^{-1} [11], it was here computed for a range of wavelengths ($\lambda = 0.5, 0.8, 1.1$ and 1.4 mm) comparable with the ones of our experimental data. The parallel with the plastronic waves is striking.

Comment 3

RC: *Fig. 5(a) presents the most important result, it would be nice to highlight it more. It would be helpful to indicate the conventional deep water capillary wave velocity on this figure – that will make Fig. 1 superfluous. The authors state that the measured phase velocity shown in Fig. 5(a) "appears to asymptotically converge towards the theoretically predicted phase speed of conventional unconstrained capillary waves." I do not see any evidence of such asymptotic convergence in the data, although it is of course expected in the limit of infinite pillar spacing. (However, the plastron will no longer exist at that limit.) Please specify which equation from Ref. 11 was used to calculate the red curve.*

Figure 3: (a) The dispersion relation of the plastronic waves with respect to the inter-pillar spacing s , with fixed pillar height $h = 53 \mu\text{m}$. Exemplary values of the ratio Ω/k (i.e., the phase speed), indicated in orange on the plot, confirm the dispersive behaviour, emphasizing as the pillar spacing decreases. (b) The phase speed as a function of the pillar spacing, with fixed pillar height $h = 53 \mu\text{m}$, all wavelengths combined. The comparison of experimental measurements is done with the theoretical phase speed of conventional (unconstrained) deep water waves ($\lambda = 1 \text{ mm}$) and with the semi-empirical model of Scott and Benjamin [11]. This model, which describes the phase speed of an interfacial wave travelling in a deep water-filled channel as a function of the wall spacing b , plotted here for different wavelengths, shows a good agreement with the experiments.

AR: We thank the Reviewer for the encouragement to highlight Fig. 5(a) results, which now is addressed in Fig. 3. There, the dispersion relation is now provided in Fig. 3(a). Besides that, Fig. 3(b) allows the comparison of the phase speed of plastronic waves to that of conventional deep water capillary waves, as suggested by the Reviewer. Accordingly, this made former Fig. 1 superfluous, so that it could be removed in the revised manuscript. The comparison is now clarified. Also the equation from Ref. [3] (formerly [11]) employed in Fig. 3(b) was Eq. (42). Because the same equation was already stated in another publication of the same two authors, one year earlier, that other reference was preferred. It is the following article by Scott and Benjamin: [11], in which there is only one equation. That equation has now been directly written on the graph itself, in Fig. 3(b), and an appropriate reference to it was added to the caption.

The following comment was added to the revised manuscript:

Similarly as previously observed in Fig. 2(c), the phase speed of plastronic waves grows with increasing wavelength, which is more obvious in Fig. 3(b). There, the phase speed of the plastronic waves is compared with (i) the theoretical phase speed of conventional (unconstrained) deep water waves, computed from Eq. (1) for $\lambda = 1$ mm, and (ii) the semi-empirical model of Scott and Benjamin [11], which describes the phase speed of an interfacial wave travelling in a deep water-filled channel.

Comment 4

RC: *The information presented in other panels in Figs. 4 and 5 is somewhat poorly organized. The way the data are presented in Fig. 4(a,b) is especially confusing. Why not show conventional $y(x)$ plots, e.g. wavelength as a function of pillar spacing at constant height and wavelength as a function of pillar height at constant spacing? Adding the acoustic intensity as another parameter adds confusion because of possible nonlinear effects. Why not limit the presentation to low intensity data? (High intensity data could be then shown in the supplement.) Fig. 5(c) is not too useful. The phase velocity of capillary waves is a local property and should not depend on the size of the plastron as long as the latter is much larger than the wavelength. The authors state that the velocity is lower for the larger plastron. I do not see any convincing evidence of that: what I see is that for the smaller plastron the spread in the data is larger, indicating a larger statistical error in the measurements.*

AR: We agree with the Reviewer's comment that the figures could be made more clear. These plots have been replaced with more conventional $y(x)$ plots, so that now the results in Fig. 2(a), 2(b), 2(c) and 3 (formerly Figs. 4 and 5) appear more clear. Fig. 2(a), 2(b) and 2(c) respectively illustrate the wave frequency vs. the AM frequency, the angular frequency vs. the wavenumber, the phase speed vs. the pillar height. Fig. 3(a) and 3(b) respectively illustrate the angular frequency vs. the wavenumber for different pillar heights and the phase speed vs. the pillar height.

To minimise possible nonlinear effects due to varying acoustic intensity, pinpointed by the Reviewer, the new series of experiments based on an AM driving signal was conducted at constant driving amplitude, as mentioned already in Comment 1. The employed driving voltage of 0.5 V amplified $50 \times$ equals 25 V, which delivered a focal acoustic pressure of 1.5 MPa, according to free-field hydrophone measurements.

The previous results with the acoustic intensity as a varying parameter were moved to SI, as mentioned before, in Fig. S7 and S8.

We agree that the results on the possible influence of the plastron volume on the wave behaviour, as presented in Fig. 5c (formerly), are inconclusive. In the absence of strong experimental evidence of the mechanisms responsible for the phase speed decrease at higher pillar height, the discussions on the plastron compressibility and elasticity, and on the plastron volume were removed. The revised manuscript now mentions this:

For higher pillars ($> 37 \mu\text{m}$), a drop in phase speed is also observed, as the pillar height varies from 37 to 71 μm . The local maximum formed around 37 μm suggests that more than one mechanism compete in the dispersion relation. The displacements of the gas-water interface induced by focused US in superhydrophobic samples can be considerable, up to 20 % of the plastron thickness, as suggested by the experimentation in superhydrophobic channels presented in SI, Fig. S3. Therefore, gas flow within the plastron and the role of the viscous boundary layer and of the gas elasticity cannot be ruled out, as possible driving mechanisms and require attention from future research to further explain the discovered phenomena.

Comment 5

RC: *Throughout the paper, the analysis is presented in the form of speculative verbal statements not supported by any model or quantitative estimates. While speculative statements are sometimes admissible, in a physics paper they should be used very sparingly. Some of these statements are clearly incorrect: or example, it is well known that the capillary wave propagation is not affected by the elasticity of the liquid or gas as long as the capillary wave velocity is much smaller than the speed of sound. The suggestion that the viscous boundary layer affects the capillary wave velocity appears to be incorrect either. It is clear that a sound physical model describing capillary waves in a plastron is currently lacking and I would recommend to acknowledge that it is lacking instead of offering unsupported explanations.*

AR: Following the advised suggestion of the Reviewer and in the absence of models or strong experimental evidences as a support, we removed the speculative statements on the influence of the gas elasticity on the wave behaviour.

We also acknowledge that the statements on the possible impact of viscous shear forces due to no-slip condition at the wall are speculative. However, knowing that theoretical models describing the case of acoustic waves or shallow water waves slowed down by the viscous boundary layer do exist, the possibility that it can play a role on the plastronic waves behaviour cannot be excluded either. Because this aspect was already discussed in detail in Comment 1, our response is here shortened.

To conclude, we agree that a sound physical model describing capillary waves in a plastron is lacking. Accordingly, this has been clarified in the manuscript, with the following lines, already mentioned in our response to Comment 1:

Nevertheless, as the physics of the exposed wave phenomena is not predicted by the existing theory, developing a new model of interfacial wave propagation in viscous shallow gas is necessary to thoroughly explain the plastronic waves behaviour.

Comment 6

RC: *The title should be descriptive. I would recommend to mention capillary waves and plastron in the title.*

AR: A new, more descriptive title has been suggested: *“Fast capillary waves on an underwater superhydrophobic surface”*.

Our suggestion is not to include the word "plastron" in the title, as it is a word specific to a scientific field, not necessarily known to the broader community. Instead, we believe that the words "superhydrophobic surface" are more familiar to collective knowledge, and hence will enhance the visibility and findability of this article.

Comment 7

RC: *The expressions such as “very first and unique” are inappropriate in a scientific paper, especially in the abstract.*

AR: We agree with this comment and have corrected it. The abstract is now the following:

*The propagation of interfacial waves in deep and in shallow water, broadly studied over centuries, is a common event, which anyone contemplating the ocean waves washing ashore could witness. As a complementary configuration, we introduce interfacial waves propagating in a **shallow gas -like condition**. This is experimentally achieved using an underwater superhydrophobic surface as the **interfacial wave** propagation medium for its ability to stabilize a well-defined microscale thin gas layer trapped between microstructures, called **plastron**. The acoustic radiation force produced with focused MHz ultrasound was used to successfully trigger kHz "plastronic waves" with unprecedented features, i.e., (i) **a remarkably high propagation speed up to 45 times faster than conventional deep water capillary waves of comparable wavelength** and (ii) **a nonlinear relation of the propagation speed with the gas layer geometry, hence deviating significantly from the behavior of capillary waves in deep and in shallow water**. Based on this and on the observed variation of wave speed over time in conditions of gas-undersaturated or -supersaturated water, their usefulness for the non-destructive monitoring of the plastron's stability and **the spontaneous** air diffusion is eventually demonstrated.*

Comment 8

RC: *In the caption to Fig. 3, please first describe panel (a), then panel (b). Please label the vertical axis.*

AR: Thank you for these observations. The label of the vertical axis had been cropped by mistake in the LaTeX layout of the figure. This has been fixed.

The caption has also been rearranged so that the panel (formerly, a) showing the acoustic field was described before the panel (formerly, b) showing an exemplary case of plastronic waves. Also, Fig. 3 (formerly) and Fig. 2 (formerly) have been merged together to make one single figure (Fig. 1) gathering all methods and setup information, as we believe that this will improve reading. Fig. 1 and its caption are now the following,

Comment 9

RC: *The caption to Fig. S4 should describe every panel in the figure.*

AR: All panels of Fig. S9 (formerly, Fig. S4), from (a) to (p) are mentioned and described in the caption. The figure and its caption are now as follows:

Figure 1: (a) The experimental setup featuring a high-speed camera (159 000 fps) allows the study of a water-submerged superhydrophobic surface placed at the focus of co-axial optical and acoustic fields. The response of the plastron to an amplitude-modulated (AM) US pulse takes the shape of interfacial perturbations travelling along the gas-water interface from the acoustic focal point, the ripples with same coherence forming patterns of concentric circles, schematised in (b). The driving signal has the form $y(t) = A \sin(2\pi f_{\text{ac}} t) \cdot \sin(2\pi t/T_{\text{AM}})$, with the constants $A = 0.5 \text{ V}$ and $f_{\text{ac}} = 2.5 \text{ MHz}$. The only variable parameter is the AM period T_{AM} , which lies in the range 100-500 μs and affects the total duration of the signal, as $t \in [0, 3T_{\text{AM}}]$. (c) An exemplary image acquired by Scanning Electron Microscopy shows the details of a superhydrophobic sample with 25 μm -high pillars, spaced apart by 25 μm . (d) The axisymmetric field of acoustic pressure produced by the HIFU transducer was measured in a free field (in the absence of a superhydrophobic surface) using a needle hydrophone. (e) The radial waves at the gas-water interface are optically assessed from the top-view refraction patterns induced by the moving gas-water interface. The waves analysis based on the tracking of the wavefronts, i.e., the wave lines of same phase, normally done in polar coordinates (Details in SI, Fig. S5 and S6), is here conceptualised on an unprocessed high-speed footage.

Figure S9: Microfabrication steps process of the superhydrophobic surfaces. The master made on a silicon wafer (a) is spin-coated with a photosensitive polymer, which is SU-8. Tuning the height of the microstructures can be done by varying the rate of spinning, details in Table S1. Following a given design (b), a selective UV exposure of SU-8 is done using a UV laser beam (c), resulting in a pattern formed in SU-8 (d) resembling to micro-pits after washing (e). A fluoropolymer is deposited on the master to later enable an easy peeling of PDMS (f). PDMS is casted on the master and put to bake (g). The hardened casting is peeled off (h). Resulting PDMS micropillars (i). PFOTS deposition on PDMS micropillar arrays (j). The empirically-defined angle of UV exposure (365 nm , 200 mJ m^{-2}) and cooking time allow to give the micropillars the wanted geometry. Images obtained by Scanning Electron Microscopy (SEM) show examples of the so-produced different heights of micropillars, which are $21\text{ }\mu\text{m}$ (k), $25\text{ }\mu\text{m}$ (l), $37\text{ }\mu\text{m}$ (m), $53\text{ }\mu\text{m}$ (n) and $71\text{ }\mu\text{m}$ (o). A top view of the array edge and the surrounding wall is also depicted (p).

Comment 10

RC: While the procedure of measuring the phase velocity is described in some details, it would be helpful to describe the procedure used to measure the wavelength and frequency.

AR: The procedure used to measure the frequency was indeed missing. The frequency measurement is based on the elapsed average time between successive wavefronts with the same phase. The operation is performed across several wavefronts, so that an average is eventually calculated. Details on the image analysis method are now provided as in the following revised Figs. S5 and S6.

Figure S5: Step-by-step image processing method. From left to right, the optical information expressed in polar coordinates (a) is sequentially de-noised (b), convoluted by a vertical kernel (c), edge-highlighted, and thresholded. The phase speed is eventually averaged from the angles of the obtained wavefronts (e). A fast Fourier transform applied on the time signals at each radial distance (f) allows to show that the wave frequency content is monochromatic (composed of one single frequency) and quickly dampens with radial distance r , following a power law $e^{\alpha(f)r}$. In this exemplary case, the frequency $f \simeq 10.4$ kHz, the wavelength $\lambda \simeq 810$ μm and the attenuation ratio $\alpha = 1.44$ dB mm^{-1} . The result suggests that the detected waves travel outwards from the point of actuation at a phase speed $c_p \simeq 12$ m s^{-1} , which is considerably faster ($> 16\times$) than the phase speed of conventional capillary waves with similar wavelength, travelling in deep water ($c_p \sim 0.75$ m s^{-1}) and in shallow water ($c_p \sim 0.46$ m s^{-1}).

Complementary information was also added on that regard in the main document with the following added lines:

The conversion of the high-speed video information into a polar coordinate system, the origin being the image-detected center of the acoustic actuation, allowed to display the waves into a time vs. radial distance arrangement, as exemplified in Fig. S5. After a detection of the wavefronts, the wave speed c_p is first evaluated, based on the slope. Then, the wave period T is obtained by computing the average time spacing between wavefronts with same phase, as shown in Fig. S6. Eventually, the wavelength λ and the wave frequency f are such that $f = 1/T$ and $\lambda = c_p/f$.

Figure S6: The conversion of the high-speed video information into a polar coordinate system, the origin being the image-detected center of the acoustic actuation, allows to re-arrange the waves into a time vs. radial distance form, as exemplified in (a). After detection of the wavefronts (b), the wave period T is obtained by computing the average time spacing between wavefronts with same phase, *e.g.*, between the pair of wavefronts numbered 1 and 3. The operation is repeated for all pairs of detected wavefronts, such as numbered i and $i+2$. (c) The wave period T is eventually the mean of all values, and the wave frequency f is such that $f = 1/T$.

Reviewer #3 (Remarks to the Author):

RC: *The paper focuses on the capillary waves generated at the surface of a plastron, showing some interesting results. The experiments are well-conducted, particularly considering their difficulty, the studied waves are highly dissipated and last only a few wavelengths. The influence of the microstructure is studied, showing a significant impact on the propagation of capillary waves due to (i) the interface confinement between pillars and (ii) the air confinement in the plastron. Although the experiments are well-described and analyzed, the analysis could benefit from some clarification in the main manuscript. However, the interpretation is somewhat confusing and requires a more detailed analysis to be publishable. Finally, the last part presents a potential application to monitor the plastron dynamics but seems a bit irrelevant due to the complexity of interpreting the results.*

In conclusion, even though the experiments are interesting and well-performed I cannot recommend publication because the interpretation and some analysis parts present several issues as listed below:

AR: We thank the Reviewer for the positive evaluation.

With a view to facilitate the interpretation of the results, a new assessment of the phenomena for sinusoidal actuation at controlled frequencies has been provided. This is elaborated more thoroughly within the response letter below.

Comment 1

RC: *The authors interpret the change in phase velocity using either the influence of the wall (spacing dependency) or of the dissipation/compressibility (height dependency). However, they overlook the influence of the wavelength on the velocity, while capillary waves are dispersive. The influence of the wavelength should be taken into account or at least discussed in the paper.*

AR: It is true that capillary waves are dispersive and behave differently as a function of the wavelength and frequency. For that reason, we agree that the wave parameters must be controlled to assess the influence of the air layer geometry only on the wave behavior, which was not the case with the ultrasound pulse as the driving signal. To better control the wave parameters, we have conducted a new set of experiments, using an amplitude modulated sine wave as the driving signal (instead of a short 20 μs pulse), in order to force the waves to oscillate at a given frequency. The new results are in line with the original results and are now presented in the revised Fig. 2(a), 2(b), 2(c), 3 and 4. The latter (Fig. 4) is shown below.

The influence of the wavelength was addressed *via* the analysis of the dispersion relation of the plastronic waves and *via* the phase speed with respect to the pillar spacing or pillar height, at given wavelength. Together with these graphical plots, the following key statements were added to the section 2.3 on the influence of the plastron thickness on the wave characteristics:

In the dispersion relation shown in Fig. 2(b), the most striking feature is that the angular frequency of the plastronic waves is about 10 to 20 times higher than that of deep water capillary waves with similar wavelength, as computed from Eq. 1 and plotted here in dashed line. This represents a significant increase of the phase speed $c_p = \Omega/k$, which is all the more emphasized, when the comparison is done with the case of shallow water waves, also represented here, in solid line. A maximum speed for this data set is reported at $\sim 11.6 \text{ m s}^{-1}$ and is associated to the greatest wavelength ($\lambda = 1220 \mu\text{m}$) and to a pillar height of $37 \mu\text{m}$. Between the different configurations of pillar height h , shifts in phase speed can be observed, and the slowest waves are associated to $h = 21 \mu\text{m}$ and $h = 71 \mu\text{m}$.

Figure 4: The observation of the relative change in the phase speed of plastronic waves with respect to time allows to monitor the gas exchange taking place between the plastron and the water phase. The plastron is actuated with successive AM pulses $y(t)$ with a given pulse repetition period (PRP). The recording is done over a duration of 4 minutes, for a total of 49 acquisitions in the case of a 5 seconds PRP (a) and 13 acquisitions in the case of a 20 seconds PRP (b). On completion of recording, the relative change in the phase speed significantly differentiates between the two explored configurations, in blue for a gas-undersaturated water and in red for a gas-supersaturated water, revealing the spontaneous outgassing and ingassing, respectively, of the superhydrophobic plastron under monitoring. A similar behaviour observed in both cases of PRP indicates that the US actuation and the propagating waves do not accelerate the gas exchange processes.

Another contrasting characteristic of the plastronic waves, visible in Fig. 2(c), is the increase in phase speed with greater wavelength, which contradicts with the behaviour of conventional capillary waves, as shown in SI, Fig. S1. For a given wavelength, the results of Fig. 2(c) evidence a quadratic-like relation of the phase speed and the pillar height with downward curve and a local maximum around $h = 37 \mu\text{m}$. Below and above that value of pillar height, a drop of phase speed is systematically observed, highlighting the action of a previously undemonstrated mechanisms slowing down the wave propagation.

As a reminder, Fig. S1 is the following:

Figure S1: Phase speed of interfacial waves in water of depth 1 m (dashed line) and 50 μm (solid line), as a function of the wavelength, in comparison with our experimental measurements of plastronic waves (red rectangle).

The following was added to section 2.4 on the influence of constraints and edges on the wave characteristics:

With constant pillar height ($h = 53 \mu\text{m}$), we now tune the interface three-phase contact density by employing various pillar spacings s : 15, 20, 25, 35, 45, 55 and $65 \mu\text{m}$. The dispersion relation of the produced plastronic waves is illustrated in Fig. 3(a). Because all these wavenumber-to-frequency linear relations have a positive y-intercept, dispersion occurs for every configurations of pillar spacing. This is confirmed on the graph by exemplary values of the ratio Ω/k , which varies with respect to k . These results thus suggest that the dispersion relation of the plastronic waves can be modulated by varying the pillar spacing and thus the density of three-phase contact line.

Similarly as previously observed in Fig. 2(c), the phase speed of plastronic waves grows with increasing wavelength, which is more obvious in Fig. 3(b). There, the phase speed of the plastronic waves is compared with (i) the theoretical phase speed of conventional (unconstrained) deep water waves, computed from Eq. (1) for $\lambda = 1 \text{ mm}$, and (ii) the semi-empirical model of Scott and Benjamin [11], which describes the phase speed of an interfacial wave travelling in a deep water-filled channel.

The adhesive forces holding the plastron follow a quadratic growth with increasing density of three-phase contact line [5]. As a result, this will make the interface appearing stiffer from the perspective of the propagating wave, and thus will increase the wave speed, which is confirmed by Fig. 3(b). Inversely, the decrease of the interfacial tensions due to a larger spacing between the solid structures (i.e., the pillars or the walls) slows down the waves. For the largest spacing, the phase speed of the plastronic waves approaches that of conventional waves in deep water, as it is expected with the depletion of edges and constraints [13]. Importantly, these experimental results are in line with the properties of capillary waves in water-filled channel, for which the propagation speed is decreasing with the broadening of the channel in which they travel [11]. Although the focus of their model is on millimeter-scale water waves with wave speed not exceeding 2 m s^{-1} [11], it was here computed for a range of wavelengths ($\lambda = 0.5, 0.8, 1.1$ and 1.4 mm) comparable with the ones of our experimental data. The parallel with the plastronic waves is striking.

Comment 2

RC: *Compressibility of the air layer is invoked to explain the slowing down of the wave with height. However, capillary waves conserve volume. One could argue that local compression might occur, but as the speed of sound is 30 times faster than the observed waves, air compressibility should not play a significant role.*

AR: *After a meticulous investigation of the literature, we came to the same conclusion as the Reviewer and agree with the comment. Because the statement on the influence of compressibility of the air layer was not supported by strong experimental evidence or by the literature, this part of the discussion was removed from the manuscript. The mechanism responsible for the slowing down of the wave with high pillars was thus left as simple descriptions of the results provided in Fig. 2(c), while specifying that further investigations are necessary to understand the driving mechanism at higher pillars. However, as we observed, in superhydrophobic channels, deformation of water-gas interface with displacement amplitude of up to 20% of the plastron height, as demonstrated in SI Fig. S3, gas flow within the plastron cannot be excluded.*

The following lines of the manuscript were revised as follows:

For higher pillars ($> 37 \mu\text{m}$), a drop in phase speed is also observed, as the pillar height varies from 37 to $71 \mu\text{m}$. The local maximum formed around $37 \mu\text{m}$ suggests that more than one mechanism compete in the dispersion relation. The displacements of the gas-water interface induced by focused US in superhydrophobic samples can be considerable, up to 20 % of the plastron thickness, as suggested by the experimentation in superhydrophobic channels presented in SI, Fig. S3. Therefore, gas flow within the plastron and the role of

the viscous boundary layer and of the gas elasticity cannot be ruled out, as possible driving mechanisms. Nevertheless, as the physics of the exposed wave phenomena is not predicted by the existing theory, developing a new model of interfacial wave propagation in viscous shallow gas is necessary to thoroughly explain the plastronic waves behaviour.

As a reminder, Fig. S3 is the following.

Figure S3: The capability of Acoustic Radiation Force to induce a mechanical perturbation on a gas-water interface is demonstrated under high-speed imaging with the side view of a gas-filled microchannel submerged in water (long axis along y-axis, channel height = 100 μm and width = 75 μm), showing travelling interfacial waves propagating along the y-axis. For the generation of the interfacial wave, the US source is ON for a duration of 20 μs with pulse-average acoustic intensity $\simeq 14.1 \text{ mW cm}^{-2}$.

Comment 3

RC: *The slowing of the phase velocity at small pillar height is interpreted by viscous dissipation, which seems feasible. However, the reference 42 used by the authors to support their claim is a conference paper that appears to study longitudinal waves in soft solid. The authors should look into the literature on viscous gravito-capillary waves. Moreover, the authors did not discuss other known sources of dissipation such as surface contamination, which could strongly affect the results over time, or meniscus dissipation due to flow in the vicinity of the pillars, which could depend on the thickness as well.*

AR: We thank the Reviewer for this comment and the suggestion. A more appropriate and cautious discussion on the relation between the experimental results with the literature was conducted, especially concerning the slowing down of the wave speed at small pillar height due to possible viscous dissipation. Speculative statements were removed from the manuscript. Instead, the revised discussion on Fig. 2(c) is now limited to simple descriptions of the results and to the statement that a slowing down of the waves is observed for the pillar heights such that $h \simeq \delta$, where δ is the thickness of the viscous boundary layer. Now, it writes:

In the context of ocean waves, the slowing down of shallow water waves with the thinning of the water depth is a known phenomenon. It has been repeatedly suggested by Lamb [9], Walbridge and Woodward [13] and Gjevik [6] that the phase velocity of an interfacial wave travelling on a medium's interface is weakly influenced by its viscosity, except, when the medium's depth approaches the thickness of the thin layer of viscous-dominated fluid forming close the solid boundary. The experimental implementation of their analytical description confirmed that the motion of the water particles driven by the swelling interface can be slowed down by this so-called viscous boundary layer, through fluid deceleration due to viscous shear stresses. A similar mechanism in the context of acoustics in narrow slit cavities was also reported more recently and likewise attributed to boundary-layer drag forces [14].

An analogous force in the case of air is expected to be present, but considerably smaller, since air is about 50 times less viscous than water [1]. In the frequency range 9.5-12.1 kHz of the plastronic waves studied as a function of the pillar height, the viscous boundary layer associated to a wave perturbation in air (kinematic viscosity $\nu \simeq 15.6 \times 10^{-6} \text{ m}^2 \text{ s}^{-1}$, at 25°C) has a thickness $\delta = \sqrt{\nu/\pi f}$ [13], approximating $21.5 \pm 1.3 \text{ }\mu\text{m}$. Because the smallest pillar heights ($h = 21$ and $25 \text{ }\mu\text{m}$) investigated in this work are such that $\delta \simeq h$, the slowing down of the plastronic waves due to viscous effects as the plastron gets thinner cannot be excluded. For higher pillars ($> 37 \text{ }\mu\text{m}$), a drop in phase speed is also observed, as the pillar height varies from 37 to 71 μm . The local maximum formed around 37 μm suggests that more than one mechanism compete in the dispersion relation. The displacements of the gas-water interface induced by focused US in superhydrophobic samples can be considerable, up to 20 % of the plastron thickness, as suggested by the experimentation in superhydrophobic channels presented in SI, Fig. S3. Therefore, gas flow within the plastron and the role of the viscous boundary layer and of the gas elasticity cannot be ruled out, as possible driving mechanisms. Nevertheless, as the physics of the exposed wave phenomena is not predicted by the existing theory, developing a new model of interfacial wave propagation in viscous shallow gas is necessary to thoroughly explain the plastronic waves behaviour.

Concerning the other known sources of contamination, the following was added to the SI:

Surface contamination was prevented as much as possible by (i) keeping the samples stored in a closed, clean box, away from light, (ii) changing the samples to new ones as often as possible, (iii) using fresh Milli-Q water in experiments, (iv) cleaning the acrylic tank after every day of experiments and (v) alternating the order of experiments with different samples. With this routine, we believe that surface contamination or bias from potential contamination could be successfully limited to a minimum, and be excluded as a mechanism responsible for the important wave attenuation presented in Fig. S8(d).

The possibility of the meniscus dissipation due to flow in the vicinity of the pillars is an interesting consideration. However, other imaging techniques, such as confocal microscopy, would be here needed to describe the meniscus motion. Due to incompatibility of confocal microscopy with high-speed imaging, necessary to visualise the fast motion of the waves (kHz oscillation of the meniscus), we believe that the consideration of meniscus dissipation is unfortunately out of scope of this work. However, this is a consideration that was added to the SI, in the section detailing the wave attenuation ratio. It reads now:

Due to incompatibility of the high speed imaging setup for observing the meniscus dynamics, the investigation of the influence of the meniscus dissipation on the wave attenuation, as suggested by Kidambi [8], is considered outside the scope of this work.

Another typical cause of wave energy dissipation is the formation of turbulent flow in the vicinity of the air-water interface. Because the Reynolds number in our configuration is expected to be low, in range 10-75, we believe that the gas motion within the plastron is laminar-dominated. Subsequently, turbulent flows are not expected in the plastron and are thus not considered as a likely contribution to the wave energy dissipation.

Comment 4

RC: *Figures 4a) and b) are unreadable. The authors should plot wavelength and frequency as functions of spacing. I understand the original idea; however, the influence of pillar height is flattened by the variation in pillar spacing, as shown in figures c) and d). Moreover, if I understand correctly, the standard deviation is based on three experiments, which seems low.*

AR: We agree with the Reviewer. To simplify the reading of Figures 4a and 4b (now Fig. 2(a), 2(b) and 2(c)), conventional $y(x)$ plots were preferred. Also, because a new series of experiments was conducted for a better control of the frequency of these dispersive waves, a new graph of the dispersion relation, frequency as a function of wavelength, has been added to the manuscript. Fig. 2(a), 2(b) and 2(c) respectively illustrate the wave frequency vs. the AM frequency, the angular frequency vs. the wavenumber (*i.e.*, dispersion relation) and the phase speed vs. the pillar height.

The number of experiments was indeed three for each acoustic intensity, with a total of 9 per pillar height (3 different acoustic intensities per pillar height). While these results were transferred to SI, as Fig. S7 and S8, the new series of experiments with AM driving was performed at constant acoustic intensity and the number of experiments was increased to 36 repeats (4 per AM frequency), in order to provide statistically valid results.

Comment 5

RC: *It is not clear how to interpret the change in contrast. The authors seem to indicate that brighter means troughs and darker translates to crests. This vision seems in opposition with a "lens" effect of the interface. This part should have been more detailed. Additionally, when looking at the supplementary data the focal area became darker at the beginning (image 7 and 8) when the surface is expected to be pushed toward the air side by radiation pressure which also appears in contradiction with the interpretation of the author.*

AR: We thank the Reviewer for this comment. Accordingly, clarification concerning the interpretation of the image contrast was added to the manuscript. Support was sought from the literature and from supporting experiments, so that this part now reads:

The bright areas of this pattern spatially correlate with the experimentally measured locations of both the acoustic focus and the first side lobe, the latter located at a radial distance $r \simeq 0.8$ mm from the acoustic

epicentre, as shown in Fig. 1(d) and labelled in Fig. 1(e). It can therefore be deduced that brighter and darker areas respectively transcribe troughs (inward excursion, i.e., toward the gas phase) and crests (outward excursion, i.e., toward the water phase) of the gas-water interface, producing a pattern of optical refraction, i.e., the consequence of transmitted light bending due to the interface deformation. This deformation of the gas-water interface is further confirmed and elaborated thanks to a side-view demonstration of plastron deformation performed in a superhydrophobic micro-channel, as discussed in Section 1.A of the SI and illustrated with experimental images in Fig. S3.

The Section 1.A of the SI was further documented as follows:

Since the direct observations of the mechanical displacement are challenging from a side view due to optical distortion caused by the micropatterned structure, the capability of ARF to perturbate a gas-water interface is exemplified in Fig. S3 in a gas-filled microchannel (width and height of 75 μm and 100 μm , respectively). For quantitative spatiotemporal study of the perturbations along the superhydrophobic surface, this work concentrates to studies with top-view perspectives of micropillared array with gas occupying the space between the pillars. However, the side-view results of Fig. S3 allowed to conclude that bright and dark optical information of the top-viewed circular plastronic waves presented in the main document respectively corresponded to concave and convex bulges of the water interface, from the camera's point of view. This follows a similar logic as in the work of Walbridge and Woodward [13], where a convex bulge of the water interface, from the camera's point of view, translates into a bright optical information. The interfacial waves, oscillating in the kHz range, start to propagate as a result of the US actuation, with characteristic time scale ranging from tens to hundreds of microseconds, which is a considerably slower event than that of the acoustic time scale oscillatory actuation (period $T_{ac} = 0.4 \mu\text{s}$ at 2.5 MHz).

Concerning the focal area becoming darker at the beginning (image 7 and 8) of the supplementary data (video), here is our interpretation. Because these two images temporally correspond to the moment when the transient 20 μs ultrasound pulse was interacting with the plastron, we interpret this darkening as the result of optical loss due to the generation of aerosols inside the plastron. This is a phenomenon that we have observed, while conducting experiments in a micro-channel visualized from a side view. While an abrupt ultrasound actuation is likely to produce aerosols, this is something that does not seem to occur when an AM driving signal is used, as the smoother, more gradual action of the ultrasound is less likely to produce these aerosols. This explains why this darkening, which was happening with the short ultrasound pulse, does not occur anymore with the AM pulse.

For that reason, this supplementary video produced with the ultrasonic pulse was removed from the SI, and instead, the data produced with an amplitude modulated pulse (where no darkening was observed) was instead uploaded as SI video. Accordingly, commenting on this darkening is no longer relevant. Yet, it will be the topic of an upcoming study, that we recently initiated, focusing on the production of aerosols inside the plastron.

Comment 6

RC: *No details are given on the acoustic calibration, although the acoustic field is shown in Fig. 3 and some acoustic properties are given in Table 1. More importantly, the intensity given by the authors results in a radiation pressure of around 9000 Pa ($p_{rad} \sim 2I/c$) for the lowest excitation, which is several times higher than the critical breakdown pressure estimated around 300 Pa or lower (using the sliding model with $\theta_a=120^\circ$ for the lowest spacing), and the Wenzel state should have been forced as detailed in Ref. 32. The authors should have commented on this discrepancy...*

AR: The model that is referred to by the Reviewer is well known to us. It involves a static driving pressure, which

is given time to achieve impalement (*i.e.*, plastron collapse). This contrasts with the present work employing a transient tapping driving pressure, which is not given time to achieve impalement, due to high level of frequency and short duration of actuation (< 1 ms). As a result, the integral of the acoustic energy with time, *i.e.*, the acoustic radiation force, employed to poke the plastron, remains small and not sufficient to induce collapse, despite the significant (1.5 MPa) amplitude of acoustic pressure employed.

On the other hand, in the case of a longer actuation with similar acoustic parameters, the plastron would indeed collapse, which we have actually observed, but considered out of the scope of this work. In fact, the work on the ultrasound-induced Cassie-to-Wenzel transition was recently published [4]. Moreover, in this recently published work, the acoustic calibration was well detailed and is very much the same as in the present work on the plastronic waves. Accordingly, although some information are provided in the main manuscript, for more details on the acoustic calibration, we refer now to this published work.

To prevent possible nonlinear effects due to employing different acoustic intensities across the different set of data, the new series of experiments was conducted at constant, low acoustic amplitude. In that way, we believe that this will facilitate the analysis and interpretation of the experimental results. Accordingly, the acoustic focal pressure employed in all the results of the main manuscript was always 1.5 MPa, which translated to an acoustic radiation force of "only" *ca.* 540 Pa. Our estimation of critical impalement pressure using the sliding model for our micropillared configuration, that is the most likely to collapse at low pressure (*i.e.*, 65 μm spacing), equals ~ 280 Pa (~ 534 Pa for sagging model). Considering that the employed driving force of 540 Pa is still above the critical breakdown pressure estimated "around 300 Pa or lower" by the Reviewer, and considering what we said above, we commented on this discrepancy mentioned by the Reviewer in the main manuscript, as follows:

As per existing theoretical models [2], a focal acoustic pressure of 1.5 MPa (measurement in a free field) is expected to be able to induce a Cassie-to-Wenzel transition of the plastron, as this pressure would correspond to a radiation force ≥ 540 Pa, which is superior than the critical impalement pressure of all plastron configurations investigated in this work. However, considering that the US actuation employed in this work is of short duration (< 1 ms) and modulated in amplitude, as detailed hereafter, the actual radiation force is small enough not to collapse the plastron. This aspect is further commented in SI, together with information on the acoustic field calibration, which is identical to that of a recent work [4].

References

- [1] F. H. Abernathy. Film notes for fundamentals of boundary layers. *National Committee for Fluid Mechanics Films*, 21623, 1970.
- [2] D. Bartolo, F. Bouamrine, E. Verneuil, A. Buguin, Silberzan, and S. Moulinet. Bouncing or sticky droplets: Impalement transitions on superhydrophobic micropatterned surfaces. *Europhysics Letters*, 74(2):299–305, 2006.
- [3] T. Brooke Benjamin and J. C. Scott. Gravity-capillary waves with edge constraints. *Journal of Fluid Mechanics*, 92(2):241–267, 1979.
- [4] A. Drago-González, M. Fauconnier, B. Karunakaran, W. S. Y. Wong, R. H. A. Ras, and H. J. Nieminen. Ultrasonic healing of plastrons. *Advanced Science*, 2403028, 2024.

- [5] A. L. Dubov, A. Mourran, M. Möller, and O. I. Vinogradova. Contact angle hysteresis on superhydrophobic stripes. *The Journal of Chemical Physics*, 141(7):074710, 2014.
- [6] B. Gjevik. Comments on “phase velocity of surface capillary-gravity waves”. *Physics of Fluids*, 15:368–370, 1972.
- [7] D. Heckerman, S. Garrett, G. A. Williams, and P. Weidman. Surface tension restoring forces on gravity waves in narrow channels. *The Physics of Fluids*, 22(12):2270–2276, 1979.
- [8] R. Kidambi. Meniscus effects on the frequency and damping of capillary-gravity waves in a brimful circular cylinder. *Wave motion*, 46(2):144–154, 2009.
- [9] H. Lamb. *Hydrodynamics*. University Press, Cambridge, 1895.
- [10] E. Monsalve, A. Maurel, V. Pagneux, and P. Petitjeans. Space-time resolved measurements of the effect of pinned contact line on the dispersion relation of water waves. *Physical Review Fluids*, 7(1):014802, 2022.
- [11] J. C. Scott and T. Brooke Benjamin. Waves in narrow channels: faster capillary waves. *Nature*, 276:803–805, 1978.
- [12] P. N. Shankar. Frequencies of gravity–capillary waves on highly curved interfaces with edge constraints. *Fluid Dynamics Research*, 39(6):457–474, 2007.
- [13] N. L. Walbridge and L. A. Woodward. Phase velocity of surface capillary-gravity waves. *Physics of Fluids*, 13(10):2461, 1970.
- [14] G. P. Ward, R. K. Lovelock, A. R. J. Murray, A. P. Hibbins, and J. R. Sambles. Boundary-layer effects on acoustic transmission through narrow slit cavities. *Physical Review Letters*, 115(044302), 2015.

Authors' Response to Reviews of

Fast capillary waves on an underwater superhydrophobic surface

Maxime Fauconnier, Bhuvaneshwari Karunakaran, Alex Drago-González, William S. Y. Wong, Robin H. A. Ras and Heikki J. Nieminen.

Nature Communications, Research Article, NCOMMS-24-02897-T

RC: Reviewers' Comment, AR: Authors' Response and text corrections

AR: We greatly appreciate the thorough and insightful feedback provided by the Reviewers. The comments and suggestions have helped us to significantly improve the manuscript. We have carefully revised the manuscript according to the suggestions provided by the Reviewers.

Below, we address each of the Reviewers' comments point-by-point.

Reviewer #2 (Remarks to the Author):

RC: *I greatly appreciate a new set of experiments conducted with an improved methodology. The presentation has also been improved. I would like to reiterate that this is an interesting study that will be of interest to a wide audience. However, I still find that the notion of “shallow gas” is misguided and not supported by the experimental results presented in the manuscript. The authors show very clearly that the main result, i.e., the high phase velocity of capillary waves in the plastron, originates from constraining the motion of the interface by the pillars. This has nothing to do with the “shallowness” of the plastron. There is indeed a weak dependence on the pillar height, therefore I agree that on the basis of the experimental data, the effect of the finite plastron thickness cannot be excluded. However, the data pertaining to the dependence on the pillar height are not nearly as clear-cut as the dependence on the pillar spacing. For example, the authors contend that “a systematic drop in phase speed is observed for the pillar heights (21 and 25 μm) matching the thickness of the viscous boundary layer inside the plastron.” However, according to Fig. 2(b), at some wave vectors the structure with $h = 25 \mu\text{m}$ yields the highest phase velocity. Furthermore, the authors suggest that capillary waves may get slowed down by viscosity of the “shallow” air layer, but the main effect they observe is a big increase in the phase velocity. The cited references pertaining to the viscosity effect (with the exception of Ref. [42], which is entirely irrelevant as it deals with acoustic waves rather than water waves) consider the viscosity of the liquid. The expectation that the air viscosity will cause an analogous effect is unfounded: whatever happens in the air has very little effect on the capillary wave propagation because of the very low density of air. The authors admit that the dependence of the phase velocity on the plastron thickness is not well understood and further research is needed. I find that the emphasis on the “shallowness” of the air layer in the abstract, introduction, and conclusion is not warranted and may even be misleading, as it may lead the reader to believe that the reported high phase velocity results from the shallow air layer.*

AR: We thank the Reviewer for the positive evaluation and for the presented critique, which we agree with.

The Reviewer depicts a very relevant point and certainly, we do not want to mislead the reader about the relation between the high phase speed and the shallowness of the gas layer. The emphasis on the shallowness has now been removed everywhere. As suggested by the Reviewer, references to the effects of shallowness on plastronic wave speed have been completely removed from the Abstract, Introduction and Conclusion. Several modifications were applied to the text, all highlighted in the manuscript, but only the most important ones are repeated here below.

The beginning of the abstract was changed to promote the influence of the restricted interface rather than the shallowness of the gas phase. It now reads:

The propagation of interfacial waves in free and constrained conditions, such as deep and shallow water, has been broadly studied over centuries. It is a common event, which anyone contemplating the ocean waves washing ashore can witness. As a complementary configuration, this work introduces waves propagating on an interface restricted by its pinning to the solid microstructures of an underwater superhydrophobic surface. The latter has the ability to stabilize a well-defined microscale gas layer, called plastron, trapped between the water and the solid phase.

In the Introduction, the paragraph emphasizing the effect of shallowness of a gas layer was removed. It now reads:

While capillary waves have been extensively studied in free conditions (e.g., deep water) [10, 1, 6, 13, 8, 3] and in constrained conditions (e.g., shallow water or narrow channel) [7, 11, 4, 14, 1, 5, 12, 9], to the best of

our knowledge, systematic studies on capillary waves at an interface constrained by its pinning to an array of microstructures are still lacking.

The title of Section 2.1 was modified: *Superhydrophobic underwater medium for interfacial waves*

Both Sections 2.3 and 2.4 investigating the influence of the pillar spacing and the pillar height, respectively, were swapped, to bring first the discussion on the pillar spacing and the influence of edge constraints (and later the discussion on the pillar height), hence putting emphasis on the main effect for the increased phase speed, i.e., the interface pinning.

Concerning Section 2.4, changes were made to focus rather on the pillar height, than on the plastron shallowness. First of all, it was done by modifying the Section title to:

Influence of the pillar height on the wave characteristics,

but also, by removing any mention of gas shallowness, and by rephrasing and shortening the main paragraph, avoiding at the same time redundancy. It now reads as:

With constant pillar spacing ($s = 25 \mu\text{m}$), we now investigate the wave behaviour with respect to the pillar height h : 21, 25, 37, 53 or $71 \mu\text{m}$. The dispersion relation in Fig. 3(a) facilitates the comparison at a given wavenumber between the plastronic waves and the conventional deep water capillary waves, again computed from Eq. 1. The angular frequency of plastronic waves is higher than that of conventional waves and seems to be weakly dependent on the pillar height. For a given wavelength (i.e., a given plotting colour), the phase speed evidences a quadratic-like relation with downward curve and a local maximum around $h = 37 \mu\text{m}$. Below and above that pivotal value of pillar height, a drop of phase speed is typically observed, suggesting undemonstrated mechanisms for slowing down the wave propagation.

The mention of shallow gas was also removed from the Conclusion. There, less emphasis was put on the effect of the pillar height, by removing the following sentence: *Interestingly, a systematic drop in phase speed is observed for the pillar heights (21 and $25 \mu\text{m}$) matching the thickness of the viscous boundary layer inside the plastron.*

The following sentence judged irrelevant, as it concerns acoustic waves, was also removed: *A similar mechanism in the context of acoustics in narrow slit cavities was also reported more recently and likewise attributed to boundary-layer drag forces [15].*

Finally, the Conclusion was shortened, avoiding unnecessary redundancy of speculative statements.

RC: *There are several other issues that need to be addressed:*

Comment 1

RC: *The abstract claims that “a nonlinear relation of the propagation speed with the gas layer geometry” is an “unprecedented feature” of plastronic capillary waves. I don’t think this claim is well founded. The phase velocity of conventional capillary waves is a nonlinear function of the depth of the liquid according to Eq. (1). I would also recommend to avoid superlatives such as “unprecedented.” It should be left to the reader to decide whether the results presented in the paper are unprecedented or not.*

AR: This is indeed a valuable suggestion. The end of the abstract now reads:

The acoustic radiation force produced with focused MHz ultrasound successfully triggers kHz “plastronic waves” with interesting features, i.e., (i) a high propagation speed up to 45 times faster than conventional

deep water capillary waves of comparable wavelength and (ii) a relation of the propagation speed with the geometry of the *microstructures*. Based on this and on the observed variation of wave speed over time in conditions of gas-undersaturated or -supersaturated water, the usefulness of the *plastronic waves* for the non-destructive monitoring of the plastron's stability and the spontaneous air diffusion is eventually demonstrated.

Comment 2

RC: *The data for $s = 25 \mu\text{m}$, $h = 53 \mu\text{m}$ shown in Fig. 2(b) don't seem to match the data for the same plastron in Fig. 3(a).*

AR: For $s = 25 \mu\text{m}$, $h = 53 \mu\text{m}$, the data shown in Fig. 2(a) (previously Fig. 2(b)) and in Fig. 3(a) are indeed two different sets of data, respectively produced in the experimental investigation of the pillar height and of the pillar spacing, done on different days, with different batches of superhydrophobic samples. However the numerical values do match, as demonstrated in the Figure below, where the data for $s = 25 \mu\text{m}$ and $h = 53 \mu\text{m}$, from Fig. 2(a) and from Fig. 3(a), are all plotted for the Reviewers on the same graph. The two data sets demonstrate an excellent agreement:

Comment 3

RC: *What does the “shallow water” curve in Fig. 2(b) show? The phase velocity of capillary waves on shallow water depends on the water depth. I would recommend removing this curve.*

AR: This is indeed a poorly relevant comparison. The curve of shallow water waves was thus removed, and the body text was adjusted to be in agreement with this modification.

Comment 4

RC: *I cannot quite make sense of the following text: “The adhesive forces holding the plastron follow a quadratic growth with increasing density of three-phase contact line [44]. As a result, this will make the interface appearing stiffer from the perspective of the propagating wave.” Why should the adhesive force between the droplet and the plastron discussed in Ref. [44] affect the capillary wave propagation? I would think that it’s the pinning of the interface by the pillars that makes it effectively stiffer.*

AR: We agree with the Reviewer and propose a different phrasing more relevant for the considered aspect:

The stiffness of the interface, augmented by its pinning to the solid structures, grows with the narrowing of the space between them, and thus with the density of edge constraints [9]. Equivalently, decreasing the spacing between the micropillars will make the interface appearing stiffer from the perspective of the propagating wave, and thus is anticipated to increase the wave speed, which is confirmed by Fig. 2(b). For the largest spacing, the phase speed of the plastronic waves approaches that of conventional waves in deep water, as it should occur with the depletion of constraints [14].

RC: *A few minor points*

Comment 5

RC: *It is obvious that the capillary wave frequency should be equal to twice the amplitude modulation frequency. Its not necessary to show 5 graphs demonstrating this in Fig. 2, one set of data would suffice.*

AR: This has been modified accordingly. Fig. 2(a) was moved to SI, while only one set of data ($h = 25 \mu\text{m}$, $s = 25 \mu\text{m}$) was kept in the main document, and integrated into Fig. 1, as a subfigure, Fig. 1(d).

Comment 6

RC: *The last two sentences in the 2nd paragraph on p. 6 are repeated in the caption to Fig. 1. This is unnecessary.*

AR: Thank you for the comment. These two sentences have been removed from the caption to Fig. 1, in the revised document. The two sentences mentioned by the Reviewer are now only present in the second paragraph of that Section. It reads:

The employed driving signal has the form $y(t) = A \sin(2\pi f_{ac}t) \cdot \sin(2\pi t/T_{AM})$, with the constants $A = 0.5 \text{ V}$ and $f_{ac} = 2.5 \text{ MHz}$. The only changing parameter in the signal is the AM period T_{AM} , which is in the range $100\text{-}500 \mu\text{s}$ and affects the total duration of the signal that lasts 3 cycles of AM, so that $t \in [0, 3T_{AM}]$.

Comment 7

RC: *On p. 13, frequencies from 3.3 to 19 kHz are referred to as “low frequencies”. For capillary waves, these frequencies aren’t low.*

AR: That is a good point. This has been modified accordingly in the revised document. It now reads:

... altogether form a meta-medium that can carry interfacial waves with frequency in the range 3.3 to 19 kHz and new characteristics, hence the name plastronic wave.

Comment 8

RC: *I would replace the word “eventually” in the beginning of the last paragraph of Conclusions by “finally.”*

AR: This has been modified accordingly in the revised document. It now reads:

Finally, the usefulness of the US-induced plasmonic waves as a non-destructive interpreter of the plasmon stability...

Reviewer #3 (Remarks to the Author):

RC: *The authors have made significant improvements by conducting new, better-calibrated experiments and enhancing the analysis with dispersion relation plots, which greatly strengthen the manuscript.*

I can now recommend publication, provided the comments listed below are addressed.

AR: We thank the Reviewer for the positive evaluation and the insightful comments, which we address below.

Comment 1

RC: *The AM modulated excitation is effective only within a certain frequency range. The authors speculate that this is due to a “resonant-like behavior” of the plastron. However, I believe it is related to the match between the spatial forcing (HIFU field) and the generated wavelength. The HIFU imposes a wavelength of approximately 0.8 mm, corresponding to a wavenumber of 7850 m^{-1} , which is the typical wavenumber excited in the study.*

AR: This is a good observation that we addressed in the first version of the manuscript (pre-review). The earliest steps of this work were conducted with a short pulse US actuation, which produced wavelengths in the range 650-850 μm , as it can still be seen in SI, Fig. S6. However, because it was decided at a later stage to improve the control on the frequency of the generated waves, the short pulse driving method was abandoned in favor of an amplitude-modulated (AM) driving force. Because the AM actuation (100-500 μs) was much longer than the pulse actuation (20 μs), the driving voltage had to be reduced by 4 to 6 times to prevent plastron collapse. As a result, the ARF generated by the side lobe was considerably weaker than before, to such an extent that the effect of the side lobe became questionable. Due to this, we decided to remove the details on the spatial frequency imposed by the HIFU's directivity, considering that the experienced wavelengths were now in the range 500 - 1700 μm .

It is true that $k = 7850\text{ m}^{-1}$ is a typical wavenumber excited in this study, but not necessarily less than e.g. 8400 m^{-1} or 6500 m^{-1} . To our understanding, we agree that a reasonable coupling must be met between the trough-to-trough distance ($\simeq 0.8\text{ mm}$) imposed by the HIFU transducer and the plastron mechanical behaviour, especially governed by interfacial tensions, in order to generate detectable plastronic waves. It is also possible that the plastronic waves with wavelength close to 0.8 mm have the highest amplitude, due to ideal coupling with the HIFU's directivity, hence our mention of "resonant-like behavior". However, considering that access to the side-view wave amplitude is not possible within the current experimental arrangement, this cannot be demonstrated. To avoid misleading the reader, the section in SI referred to by the Reviewer was modified accordingly, so it now reads:

Together, this allowed to study the waves behaviour as a function of the plastron geometry, as independently as possible from the other experimental parameters, such as the wave amplitude. We believe this is the case, to a limited extent. It is possible that the distance imposed by the main lobe and the first side lobe of the HIFU transducer's directivity ($\simeq 0.8\text{ mm}$), as shown in Fig. 1(e), promotes the plastronic waves with wavelength matching that distance. In the earliest steps of this work, by generating the plastronic waves with a short US pulse (20 μs), it interestingly appeared that their wavelength was in the range 650-850 μm , as shown in Fig. S6(a), therefore nearing that 0.8 mm distance. The associated ranges of wave frequency and phase speed are given in Fig. S5 and in Fig. S7(a-b). At a later stage of this work, because a better control of the plastronic waves frequency was desirable to facilitate the results analysis at a given frequency, a driving force modulated in amplitude with a given frequency was preferred over the short pulse configuration. The detectable plastronic waves so-produced by ARF with controlled frequency had a wavelength in the range

500 - 1700 μm . Within this range, it is possible that these waves generated at constant driving voltage do not have a constant amplitude across different wavelength. However, considering that an access to the side-view wave amplitude is not conceivable within the current experimental arrangement, this cannot be confirmed. The possible variation of wave amplitude was therefore disregarded across all the results with varying wavenumber and plastron geometry.

Comment 2

RC: *Please remove the gas elasticity hypothesis. As both reviewers have already explained, this is highly unlikely since the plastronic waves are slower than the speed of sound in air, meaning air compressibility cannot play a role. While the gas flow might indeed depend on the air thickness, as suggested by the authors, it will be governed by hydrodynamic laws without significant density change.*

AR: This is a mistake from our side. This should have been removed, as agreed during the first round of review. This now reads:

Because the smallest pillar heights ($h = 21$ and $25 \mu\text{m}$) investigated in this work are such that $\delta \simeq h$, the slowing down of the plastronic waves due to viscous effects as the plastron gets thinner cannot be excluded as a participating mechanism influencing the wave characteristics.

Comment 3

RC: *The colors in Figures 2c and 3b are inverted; please harmonize them.*

AR: These two figures have been harmonized accordingly. Thank you for spotting this mistake.

Comment 4

RC: *The paragraph discussing the speed of plastronic waves at the beginning of page 11 is somewhat out of context. It is not a race. Additionally, some sentences are repeated in the conclusion. I suggest removing this paragraph.*

AR: Thank you for the comment. The repeated sentences have been removed from the conclusion. The mentioned paragraph was however not completely removed, but instead dimmed and modified in consideration of the comment of the Reviewer. It was done so to also integrate better in the revised manuscript the swap of Section 2.3 and 2.4, aiming at emphasizing the results on the changing density of interface pinning (pillar spacing). The mentioned paragraph now reads:

The phase speed of plastronic waves grows with increasing wavelength, which is more obvious in Fig. 2(b). This contradicts with the behaviour of conventional deep water capillary waves, as shown in SI, Fig. S1. Another contrasting feature of the plastronic waves is that they travel considerably faster than deep water waves, as computed from Eq. 1, for $\lambda = 1 \text{ mm}$. This represents a speed-up factor of up to 45, which seems to be facilitated by the plastron configuration. Among all our experiments, the highest phase speed achieved is 22.5 m s^{-1} , which is experienced by a 13.1 kHz plastronic wave with wavelength 1.7 mm , in the case of the smallest pillar spacing $s = 15 \mu\text{m}$ ($h = 53 \mu\text{m}$).

Comment 5

RC: *Although radiation forces are a mean effect, they act on a timescale of a few acoustic periods. The fact that the transition is not observed likely has more to do with the time it takes for the interface to move within*

the microstructure, which should be limited by liquid inertia, air flow, or dissipation at the contact line.

AR: We share this understanding and we modified the text accordingly. The paragraph now reads:

However, considering that the US actuation employed in this work is of short duration (< 1 ms) and modulated in amplitude, as detailed hereafter, it might not provide sufficient time for the inertia-driven interface dynamics to locally generate plastron collapse. This aspect is briefly commented in SI, Section 1.A.

The associated part in the SI, Section 1.A reads as follows:

This paragraph comments on the absence of Cassie-to-Wenzel transition reported in this work, while the employed acoustic pressures are superior than the theoretical critical impalement pressure of the investigated plastrons [2]. While the existing theoretical models consider a static pressure, the acoustic forcing used here is of short duration (< 1 ms) and modulated in amplitude. Possibly, this does not provide sufficient time for the plastron collapse to occur through interface touchdown or sliding of the contact line during a single period of plastronic wave, which are the typically reported mechanisms of wetting [2]. We confirmed experimentally (unpublished data) that a longer actuation with same acoustic parameters can easily collapse the plastron, suggesting that the plastron dynamics is restrained by hydrodynamics phenomena, such as liquid inertia, air flow or dissipation at the contact line, with slow timescale compared to the duration of the US actuation.

Comment 6

RC: *I didn't notice this in the first round, but "plastronic wave" is catchy—I like it!*

AR: Thank you very much for your support. We do like it as well, and hope that this will be appreciated by the scientific community too.

Thank you for your feedback and valuable comments.

References

- [1] G. B. Airy. *Encyclopedia Metropolitana*, volume 5, chapter On tides and waves, page 289. Dover Publications, London, 1841.
- [2] D. Bartolo, F. Bouamrine, E. Verneuil, A. Buguin, Silberzan, and S. Moulinet. Bouncing or sticky droplets: Impalement transitions on superhydrophobic micropatterned surfaces. *Europhysics Letters*, 74(2):299–305, 2006.
- [3] G. D. Crapper. An exact solution for progressive capillary waves of arbitrary amplitude. *Journal of Fluid Mechanics*, 2(6):532–540, 1957.
- [4] F. Gerstner. Theorie der wellen. *Annalen der Physik*, 32(8):412–445, 1809.
- [5] D. Heckerman, S. Garrett, G. A. Williams, and P. Weidman. Surface tension restoring forces on gravity waves in narrow channels. *The Physics of Fluids*, 22(12):2270–2276, 1979.
- [6] H. Lamb. *Hydrodynamics*. University Press, Cambridge, 1895.

- [7] M. S. Longuet-Higgins and R. W. Stewart. Radiation stresses in water waves; a physical discussion, with applications. *Deep Sea Research and Oceanographic Abstracts*, 11(4):529–562, 1964.
- [8] J. H. Michell. On the highest waves in water. *The Philosophical Magazine*, 36(5):430–437, 1893.
- [9] E. Monsalve, A. Maurel, V. Pagneux, and P. Petitjeans. Space-time resolved measurements of the effect of pinned contact line on the dispersion relation of water waves. *Physical Review Fluids*, 7(1):014802, 2022.
- [10] J. S. Russell. Report on waves. In *Report of the Fourteenth Meeting of the British Association for the Advancement of Science*, London, 1844. John Murray.
- [11] J. C. Scott and T. Brooke Benjamin. Waves in narrow channels: faster capillary waves. *Nature*, 276:803–805, 1978.
- [12] P. N. Shankar. Frequencies of gravity–capillary waves on highly curved interfaces with edge constraints. *Fluid Dynamics Research*, 39(6):457–474, 2007.
- [13] G. G. Stokes. On the theory of oscillatory waves. *Transactions of the Cambridge Philosophical Society*, 8:441–455, 1847.
- [14] N. L. Walbridge and L. A. Woodward. Phase velocity of surface capillary-gravity waves. *Physics of Fluids*, 13(10):2461, 1970.
- [15] G. P. Ward, R. K. Lovelock, A. R. J. Murray, A. P. Hibbins, and J. R. Sambles. Boundary-layer effects on acoustic transmission through narrow slit cavities. *Physical Review Letters*, 115(044302), 2015.

Authors' Response to Reviews of

Fast capillary waves on an underwater superhydrophobic surface

Maxime Fauconnier, Bhuvaneshwari Karunakaran, Alex Drago-González, William S. Y. Wong, Robin H. A. Ras and Heikki J. Nieminen.

Nature Communications, Research Article, NCOMMS-24-02897-T

RC: Reviewers' Comment, AR: Authors' Response and text corrections

AR: We appreciate the positive feedback and comments provided by the Reviewers. This has helped us to improve the manuscript and propose a revised version.

In what follows, we address the Reviewers' comments.

Reviewer #2 (Remarks to the Author):

RC: *I appreciate the revisions made by the authors. The manuscript has been sufficiently improved to recommend publication. I have a few optional recommendations that the authors may want to consider.*

AR: We thank the Reviewer for the positive evaluation and for the recommendations.

Comment 1

RC: *In Fig. 2(a), it looks as if the beam from the light source passes through the HIFU transducer. It would be helpful to clarify this point.*

AR: The light beam indeed passes through the HIFU transducer, which presents a hole at its center. This hole, initially meant for fitting a passive cavitation probe, confocal with the HIFU transducer, facilitated here the visualization of the superhydrophobic surface from a top-view perspective. This was clarified in Section Methods of the revised manuscript, which now reads:

The visualisation is done in a top-view perspective, through the central opening of the transducer, with a 5× magnification objective (Canon Inc., MP-E 65 mm, Japan), resulting in an image scaling of 5.7 $\mu\text{m pixel}^{-1}$.

Details on the ultrasound transducer's geometrical and acoustic properties were also added to the Section Methods. It reads:

The geometrical and acoustic properties of the HIFU transducer, as described by the manufacturer, are given in what follows. The transducer with central frequency 2.5 MHz has an outer diameter of 60 mm and an inner diameter (central opening) of 22.6 mm. The curvature radius at radiating surface is 50 mm, and the focal depth is 39 mm. The pressure focal gain is 91.17, assuming 1 at the radiating surface and in a linear homogeneous field. The focal width and length at half-amplitude (-6 dB) equal 0.51 mm and 3.28 mm.

Comment 2

RC: *What is SHS in Fig. 2(a)?*

AR: Thank you for raising this important detail. The acronym "SHS" used to stand for "superhydrophobic surface" in previous versions of the manuscript, but it is no longer the case. It is thus our mistake if the acronym "SHS" was left in Fig. 2(a), without further information. Fig. 2(a) was revised, by changing "SHS" to "superhydrophobic surface".

Comment 3

RC: *It would be helpful to present a single-frame image (perhaps in the SI) showing circular capillary waves.*

AR: This is a good suggestion, although we believe that a considerable number of images on a printed format might be less self-explanatory than the video (movie #1) we already provide as a Supplementary Information.

Comment 4

RC: *P. 2: "Interfacial waves on liquids are generally categorised, according to their wavelength into two distinct groups, capillary ($\lambda \leq 1.7$ cm) and gravity ($\lambda \geq 1.7$ cm) waves [...]" Since the length 1.7 cm is specific to pure water and will be different for other liquids, consider replacing "liquids" by "water".*

AR: Thank you for this relevant suggestion. We replaced "liquids" by "water" in the revised manuscript.

Comment 5

RC: *P. 4: “This essentially results from the acoustic waves having sufficient time-averaged energy density to provide an ARF overcoming the resisting forces arising from the local interfacial tensions.” This sentence can be incorrectly interpreted to imply that the ultrasonic power should overcome a certain threshold in order to excite capillary waves. Please consider revising.*

AR: We agree with the Reviewer. To avoid wrong interpretation, the sentence was modified. It now reads:

This essentially results from the acoustic waves having sufficient time-averaged energy density to provide an ARF able to deform the gas-water interface [1].

Comment 6

RC: *P. 7: “side-view demonstration of plastron deformation performed in a superhydrophobic microchannel.” Do I understand it correctly that the microchannel shown in Fig. S3 is not a plastron? Then the expression “demonstration of plastron deformation” is inaccurate.*

AR: The microchannel shown in Fig. S3 is actually a plastron, in the defined sense that a plastron is a gas layer trapped between water and the superhydrophobic, solid features. However, unlike the array of micropillars mostly investigated in the manuscript, the plastron visualized in Fig. S3 is confined on the sides by two vertical parallel walls, hence the designation "microchannel". Because this microchannel configuration facilitated the side-view visualization of the plastron, we believe that it is a convincing demonstration of plastron deformation and propagating capillary waves.

Comment 7

RC: *Placing equations in figures as in Fig. 2(b) is somewhat unusual. It may be more appropriate to present this equation in the text.*

AR: The equation of phase speed was removed from Fig. 2(b) and defined in the main text. This part of the manuscript now reads:

Fig. 2(b) also depicts the semi-empirical model of Scott and Benjamin [3], which describes the phase speed c_p of an interfacial wave travelling in a deep water-filled channel, as follows

$$c_p^2 = \frac{1.2 (g + k^2 \sigma / \rho) + 12 \sigma / (\rho b_c^2)}{k [\coth(kh_c) + 0.0305(kb_c) - 0.000376(kb_c)^3]}, \quad (1)$$

where h_c and b_c are the walls height and spacing. The stiffness of the interface, (...)

Comment 8

RC: *P. 8: “depletion of constraints.” I would recommend to replace “depletion” by “removal”.*

AR: The revised manuscript was modified accordingly. The sentence now reads:

For the largest spacing, the phase speed of the plastronic waves approaches that of conventional waves in

deep water, as it should occur with the removal of constraints [4].

Comment 9

RC: *P. 8: “the model of Scott and Benjamin focuses on millimeter-scale water waves.” Did the authors mean to say “centimeter-scale”?*

AR: The wavelengths experimentally reported by Scott and Benjamin [3] are in the range 20-400 mm. We agree that "centimeter-scale" is thus more appropriate. This was modified accordingly in the revised manuscript.

Comment 10

RC: *P. 12: “further investigations will be necessary to understand the exact mechanisms driving the plastronic waves.” I believe in the experiment described in the manuscript the plastronic waves are driven by the acoustic radiation pressure and this is well understood? What is not understood is the dependence of the phase velocity on the pillar height.*

AR: We agree that this sentence might be misleading. By this sentence, we referred to the mechanisms "governing" the waves, rather than the mechanisms "generating" the waves. To avoid misinterpretation, the sentence was revised. The end of that paragraph now reads:

*While the impact of increasing density of the three-phase contact line on the wave speed follows known trends [3], the quadratic-like association between the phase speed and the pillar height suggests that mechanisms not predicted by previous studies compete in the dispersion relation, at the investigated wave frequencies. Although a tentative analogy was done with the literature on gravity-capillary waves slowing down due to viscous forces [4, 2], further investigations will be necessary to **explain thoroughly the experimental results reported here.***

Reviewer #3 (Remarks to the Author):

RC: *I was initially inclined to recommend publication and the new version comfort this. I have only one minor comment. I believe n in figure 4 is not defined (number of repetition ?).*

AR: We thank the Reviewer for the positive evaluation.

The parameter n in Fig. 4 was indeed not defined. It indeed referred to the number of repetitions. This is now clarified in the revised caption, the last sentence of which now reads:

The parameter n refers to the number of repetitions for each experimental configuration.

References

- [1] B.-T. Chu and R. E. Apfel. Acoustic radiation pressure produced by a beam of sound. *The Journal of the Acoustical Society of America*, 72(6):1673–1687, 1982.
- [2] B. Gjevik. Comments on “phase velocity of surface capillary-gravity waves”. *Physics of Fluids*, 15:368–370, 1972.
- [3] J. C. Scott and T. Brooke Benjamin. Waves in narrow channels: faster capillary waves. *Nature*, 276:803–805, 1978.
- [4] N. L. Walbridge and L. A. Woodward. Phase velocity of surface capillary-gravity waves. *Physics of Fluids*, 13(10):2461, 1970.